# Structural mechanisms of autoinhibition and substrate recognition by the ubiquitin ligase HACE1

Jonas Düring [1,9], Madita Wolter [1,9], Julia J. Toplak[1], Camilo Torres[1], Olexandr Dybkov [2], Thornton J. Fokkens[1], Katherine E. Bohnsack [3], Henning Urlaub [2,4,5], Wieland Steinchen[6,7], Christian Dienemann [8] & Sonja Lorenz [1]✉

Ubiquitin ligases (E3s) are pivotal specificity determinants in the ubiquitin system by selecting substrates and decorating them with distinct ubiquitin signals. However, structure determination of the underlying, specific E3-substrate complexes has proven challenging owing to their transient nature. In particular, it is incompletely understood how members of the catalytic cysteine-driven class of HECT-type ligases (HECTs) position substrate proteins for modification. Here, we report a cryogenic electron microscopy (cryo-EM) structure of the full-length human HECT HACE1, along with solution-based conformational analyses by small-angle X-ray scattering and hydrogen–deuterium exchange mass spectrometry. Structure-based functional analyses in vitro and in cells reveal that the activity of HACE1 is stringently regulated by dimerization-induced autoinhibition. The inhibition occurs at the first step of the catalytic cycle and is thus substrate-independent. We use mechanism-based chemical crosslinking to reconstitute a complex of activated, monomeric HACE1 with its major substrate, RAC1, determine its structure by cryo-EM and validate the binding mode by solution-based analyses. Our findings explain how HACE1 achieves selectivity in ubiquitinating the active, GTP-loaded state of RAC1 and establish a framework for interpreting mutational alterations of the HACE1–RAC1 interplay in disease. More broadly, this work illuminates central unexplored aspects in the architecture, conformational dynamics, regulation and specificity of full-length HECTs.

The ubiquitin (Ub) system orchestrates myriad cellular pathways through dynamic modifications of tens of thousands of sites[1]. This astounding versatility largely relies on E3s that select substrates and modify them with distinct Ub signals. E3s act downstream of Ub-activating (E1) and Ub-conjugating enzymes (E2s) and in concert with deubiquitinases. However, E3s are the most diversified enzyme class of this catalytic cascade and thus are crucial for its specificity and regulation. Consequently, E3 dysregulation is linked to various human diseases, rendering these enzymes attractive therapeutic targets.

How E3s recognize substrates is key to understanding and manipulating their activities but difficult to study structurally, as the underlying interactions are transient. Selective crosslinking enabled first views of E3 complexes with substrate-derived peptides (for example, refs. 2–6), substrate proteins or domains thereof (for example,

refs. 7–13). Most of these complexes contain RING-type E3s that facilitate direct Ub transfer from E2 to substrate[14]. By contrast, HECTs form a thioester-linked intermediate with Ub before transferring it to a substrate[15]. To drive this two-step reaction, the carboxy-terminal catalytic HECT domain transitions between different states. For Ub transfer from E2 to E3, the HECT domain lobes adopt an 'inverted-T' conformation[16,17]; subsequent Ub transfer to a substrate requires an 'L' conformation[18]. Substrate recognition by HECTs typically occurs through amino-terminal regions flanking the conserved HECT domain. Diverse, E3-specific substrate-binding motifs have been identified, for example, the WW domains of NEDD4-subfamily E3s[19], the BH3 and WWE domains of HUWE1 (refs. 20–22) and the LxxLL-motif of UBE3A[23]. How these motifs orient substrates toward the catalytic center of the full-length ligases is unclear. Recently, tetrameric UBR5 was found to use the HECT domain to sandwich substrates between its subunits[10,11]. Yet for other HECTs, the catalytic domain is insufficient for substrate recruitment and the significance of oligomerization is widely unknown. We set out to illuminate the mechanisms of substrate recognition and regulation in the structurally uncharacterized human HECT HACE1.

HACE1 is implicated in redox homeostasis[24–26] and membrane dynamics, including cell adhesion[27–29], autophagy[30,31] and Golgi turnover[32]. HACE1 expression was reported to confer protection against hemodynamic stress in the heart[31] and tumorigenesis[33–35], while loss-of-function mutations or depletion of *HACE1* lead to neurodevelopmental deficiencies in frogs and humans[25,36–39]. In several settings, phenotypes were linked with loss of the catalytic activity of HACE1 (refs. 24,26,27,30,33,40). The pathways mediating the diverse roles of HACE1, however, are probably system-dependent and are incompletely understood. Few HACE1 substrates have been identified, including the selective autophagy receptor optineurin (OPTN)[30], the Golgi t-SNARE syntaxin 5 (ref. 41) and the TNFR1 adaptor TRAF2 (ref. 40). The most established substrate is the multifunctional small GTPase RAC1 (refs. 24,27–29,42). Different Ub signals were detected on HACE1 substrates[27,30,40,41,43], yet how the ligase assembles these modifications and which determinants confer specificity has not been studied structurally. To understand the mechanisms of substrate recognition by HACE1, we focused on its interaction with RAC1.

A RHO-family GTPase, RAC1 cycles between inactive, GDP-bound and active, GTP-bound forms, allowing for spatiotemporal regulation of its interactions and functions. GTP-loading triggers conformational changes in the switch-I and switch-II regions of the GTPase fold that enable the selective engagement of effectors and regulators[44]. HACE1 specifically modifies GTP-bound RAC1 with degradative Lys48-linked Ub chains[24,27,29], thereby restricting RAC1-dependent cellular functions in various contexts. In the same vein, neuropathologic features of SPPRS (spastic paraplegia and psychomotor retardation with or without seizures) upon HACE1 deficiency are accompanied by elevated levels of active RAC1 (ref. 37). Moreover, the tumor suppressive functions of HACE1 in lung cancer were linked to reduced RAC-family GTPase activities[35].

Here, we combine cryo-EM, small-angle X-ray scattering (SAXS), hydrogen–deuterium exchange (HDX) mass spectrometry (MS), selective crosslinking and functional analyses to uncover the mechanisms of dimerization-induced autoinhibition of full-length HACE1 and selective recognition of GTP-loaded RAC1 by monomeric HACE1. Our findings have important implications for understanding the substrate specificity of HACE1 and disease-associated perturbations of its interplay with RAC1. Finally, this work unveils basic principles in the regulation and architecture of full-length HECTs, of which only few have been structurally characterized[10,11,45–51].

## Results

### Cryo-EM structure of HACE1 reveals a yin–yang-like dimer

Analytical size-exclusion chromatography (SEC) and mass photometry analyses show that purified full-length HACE1 ('HACE1 FL'; 102 kDa) is dimeric, even at low-nanomolar concentrations (Fig. 1a,b, Extended Data Table 1 and Extended Data Fig. 1a). The dimerization is observed independently of the expression system used for recombinant protein preparation (Extended Data Fig. 1b). We determined a cryo-EM structure of the HACE1 dimer at 4.7 Å resolution (Table 1, Fig. 1c,d and Extended Data Fig. 1c–g), using AlphaFold2 (AF2)-based predictions as a starting point for model building (Supplementary Fig. 1a,b). Each subunit of the dimer consists of a C-terminal HECT domain, flanked by an α-helical 'middle domain' (MID), seven ankyrin repeats (ANKs) and an N-terminal α-helix ('N-helix'; residues 1–21). Two flexible insertions within the MID ('loop 1' and 'loop 2') are not visible in the map. The MID and ANKs form a concave platform, above which the HECT domain leans peripherally. The subunits are arranged in a closed, head-to-tail fashion, giving rise to a ring made up of two sets of ANKs, MIDs and small wings of the HECT N-lobes (for HECT domain architecture, see Supplementary Fig. 1c). The large wings of the N-lobes are juxtaposed on the inside of the ring, giving rise to a yin–yang-like overall arrangement. The catalytic C-lobes protrude above the platforms, with one adopting an inverted-T conformation (molecule A). The second C-lobe (molecule B) could not be modeled, indicating inter-lobe flexibility within the catalytic domain—a functionally important property of HECTs[15,52]. Moreover, six C-terminal residues of the modeled C-lobe are disordered, in line with other structures of inverted-T-shaped HECT domains[15]. Aside from the C-lobe, the subunits of the dimer are similar to each other, as are their interfaces. Notably, AF2 predictions yield an overall similar but more compact arrangement compared to the experimental structure, resulting from inter-domain tilts within the subunits (Supplementary Fig. 1d). In the cryo-EM structure, dimerization is largely mediated by contacts of the N-helix of one subunit with the small wing of the HECT N-lobe of the other, with minor contributions from the N-terminal ANK. Consistently, a HACE1 variant lacking the N-helix ('HACE1 ΔN'; residues 22–909) is monomeric (Fig. 1a,b, Extended Data Table 1 and Extended Data Fig. 1a). The corresponding mass photometry profile shows a slight shoulder that may indicate residual, weak associations, but those are lost at reduced ionic strength (Extended Data Fig. 1h). Structure refinements with 3DFlex[53] reveal dynamic inter-domain tilting within the dimer (Supplementary Video 1). These motions—together with the orientational bias of the particles on the cryo-EM grids—have probably limited the resolution of our structure.

### The N-helix controls the conformation of HACE1 in solution

Experimental SAXS data of HACE1 FL show excellent agreement with simulations based on the dimeric structure, upon manual modeling of the second C-lobe and automated modeling of loops with AllosMod-FoXS[54,55] (Fig. 1e). This is also reflected by the experimentally derived and calculated radii of gyration being identical within error (Fig. 1f and Extended Data Fig. 2a). Moreover, SAXS-based ab-initio reconstructions of HACE1 FL with GASBOR[56] recapitulate the ring-like shape and dimensions of the cryo-EM structure (Extended Data Fig. 2b). We next compared SAXS data of monomeric HACE1 ΔN to simulations, based on a HACE1 molecule extracted from the dimer (molecule A; residues 22–903) and automated modeling of missing regions. In contrast to the dimer, a moderate fit between experiment and simulation was only achieved by multi-state modeling with MultiFoXS[54], allowing for inter-lobe flexibility within the HECT domain and inter-domain flexibility between the HECT domain and MID (Fig. 1f and Extended Data Fig. 2c,d). This suggests that monomeric HACE1 has considerable conformational freedom. Consistently, the experimentally determined maximal dimension of HACE1 ΔN is slightly larger than that of the FL dimer (Fig. 1f and Extended Data Fig. 2e). Although it cannot be excluded that HACE1 ΔN transiently associates to a minor degree, thus deviating from a rigid monomer, we consider it unlikely, as the SAXS measurements were conducted at low ionic strength (see Extended Data Fig. 1h) and with online-SEC. Our data thus argue for enhanced

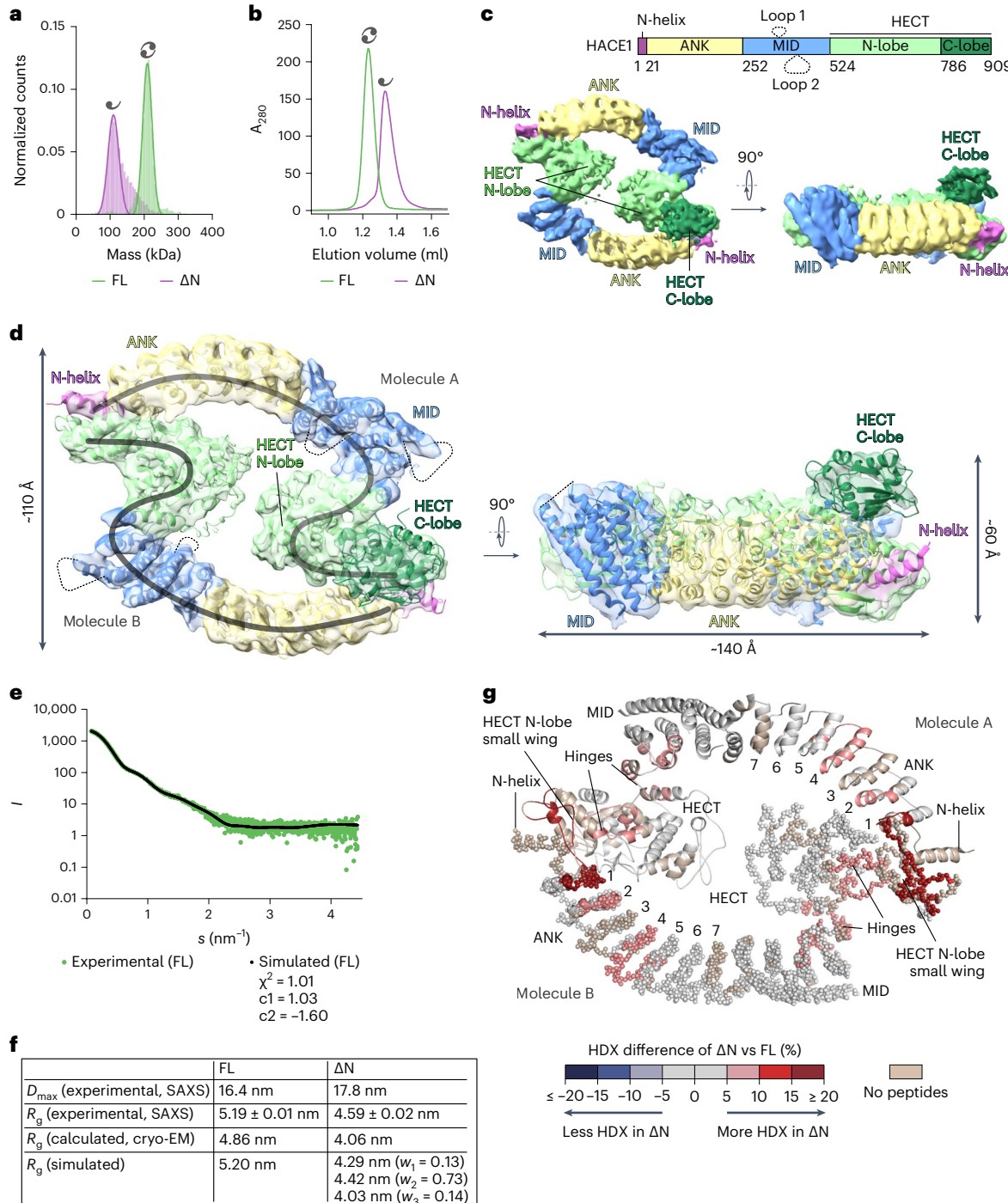

**Fig. 1 | Dimerization of HACE1 requires the N-helix. a**, Mass photometry (MP) analysis of HACE1 FL and ΔN. For molecular weights (MWs) and other parameters, see Extended Data Table 1. The symbols denote a HACE1 monomer and dimer, respectively. **b**, SEC analysis of HACE1 FL and ΔN. For MW estimates, see Extended Data Fig. 1a. **c**, Domain architecture of HACE1 (top). Cryo-EM map of HACE1 FL in two orientations (bottom). **d**, Composite cryo-EM map and structure of HACE1 FL in two orientations. Domains of molecule A are labeled. Disordered loops are marked by dashed lines; subunits are marked by curved lines. Approximate particle dimensions are indicated. **e**, SAXS data of HACE1 FL (*I*, scattering intensity; *s*, momentum transfer) (green), superposed with a simulation (black), generated with AllosMod-FoXS[54,55] based on the cryo-EM structure. The fit parameters are provided below, and the radii of gyration ($R_g$) values are shown in **f** and Extended Data Fig. 2a. **f**, SAXS-derived, calculated (based on the

cryo-EM structure) and simulated dimensions of HACE1 FL and ΔN. $D_{max}$, maximal dimension. The simulated values were generated with AllosMod-FoXS (FL) and MultiFoXS[54] (ΔN). The MultiFoXS output includes three states, for which individual $R_g$-values and occupancies (*w*) are listed. The calculated values are smaller, as they exclude flexible regions not modeled in the cryo-EM structure. **g**, HDX differences between HACE1 FL and ΔN, mapped onto the cryo-EM structure. Regions not present in both constructs or not covered are colored beige. The colors reflect changes upon dimer disruption; no rigidification (blue) is seen. The hinges between the MID and HECT domains and between the wings of the N-lobes are marked. Molecule A is shown as a cartoon and molecule B as spheres. The ANKs are numbered. The orientation of the structure is flipped compared to **d**. For related data, see Supplementary Figs. 2 and 3 and Supplementary Data 1.

**Table 1 | Cryo-EM data collection, refinement and validation statistics**

| | HACE1 FL dimer (EMD-17994), (PDB 8PWL) | HACE1 ΔN–RAC1 Q61L (EMD-18056), (PDB 8QON) |
|---|---|---|
| **Data collection and processing** | | |
| Magnification | ×105,000 | ×105,000 |
| Voltage (kV) | 300 | 300 |
| Electron exposure (e⁻/Å²) | 40, 40 frames | 60, 60 frames |
| Defocus range (μm) | 0.3–3.2 | 0.6–3.1 |
| Pixel size (Å) | 0.834 | 0.834 |
| Symmetry imposed | C1 | C1 |
| Initial particle images (no.) | 3,639,240 | 7,042,271 |
| Final particle images (no.) | 118,791 | 256,595 |
| Map resolution (Å) | 4.7 | 4.2 |
| FSC threshold | 0.143 | 0.143 |
| Map resolution range (Å) | 3.7–52.8 | 3.5–7 |
| **Refinement** | | |
| Initial model used | AF2 | AF2 |
| Model resolution (Å) | 4.7   7.6 | 4.4   6.9 |
| FSC threshold | 0.143   0.5 | 0.143   0.5 |
| Map sharpening $B$ factor (Å²) | 309.3 | Local resolution map |
| Model composition | | |
| Non-hydrogen atoms | 12,266 | 8,008 |
| Protein residues | 1,541 | 1,009 |
| Ligands | – | GTP, SIA |
| $B$ factors (Å²) min / max / mean | | |
| Protein | 30.0 / 946.7 / 185.9 | 224.6 / 1063.4 / 373.32 |
| Ligand | – | 20.0 / 20.0 / 20.3 |
| R.m.s. deviations | | |
| Bond lengths (Å) | 0.002 (0) | 0.002 (0) |
| Bond angles (°) | 0.509 (0) | 0.518 (0) |
| **Validation** | | |
| MolProbity score | 1.29 | 1.25 |
| Clashscore | 5.24 | 4.77 |
| Poor rotamers (%) | 0 (Phenix) / 0.2 (PDB) | 0 (Phenix) / 0.1 (PDB) |
| Ramachandran plot | | |
| Favored (%) | 97.97 | 98.4 |
| Allowed (%) | 2.03 | 1.6 |
| Disallowed (%) | 0 | 0 |

conformational flexibility of the HACE1 ΔN monomer compared to the FL dimer.

In an orthogonal approach, we interrogated oligomerization-induced conformational changes of HACE1 by HDX–MS (Fig. 1g and Supplementary Figs. 2 and 3). Compared to the FL dimer, HACE1 ΔN displays locally elevated HDX, indicating enhanced exposure to solvent and/or flexibility of the monomeric state. The strongest effects are seen in the N-terminal ANK flanking the critical N-helix and in the small wing of the HECT N-lobe. Additional changes affect the N-terminal half of the ANKs adjacent to the dimerization site and the

hinges between the middle and HECT domain and between the two wings of the HECT N-lobe, respectively. The HDX pattern thus corroborates the dimeric arrangement seen by cryo-EM and illustrates how dimerization-dependent effects are propagated from the subunit interface through the ligase fold.

Taken together, these integrated analyses demonstrate that purified HACE1 FL adopts a dimeric state in solution that critically depends on the N-helix and closely resembles the cryo-EM structure. The conformation of HACE1 ΔN can be recapitulated by an ensemble of monomers, as extracted from the dimer structure, reflecting inherent flexibility due to inter-domain hinges.

**Dimerization of HACE1 confers autoinhibition**

To understand the consequences of dimerization for HACE1 activity and the importance of the N-helix, we generated full-length HACE1 variants in which three leucine residues in the hydrophobic face of the amphipathic N-helix were individually substituted by aspartate (Fig. 2a). Each mutation shifts the conformation of HACE1 toward a monomer, albeit to different degrees: the L11D and L15D substitutions have stronger effects than L8D (Fig. 2b and Extended Data Table 1), mirroring the extent by which the altered residues engage in inter-subunit contacts. Leu11 and Leu15 are in direct proximity to hydrophobic residues in the adjacent N-lobe, including Trp693, Ile694, Leu704 and Leu706, whereas Leu8 is located more peripherally. In HDX–MS analyses, the L11D and L15D substitutions recapitulate the effects of the ΔN truncation (Extended Data Fig. 3). The HACE1 dimer can thus be efficiently disrupted by individual point mutations.

We next performed reconstituted multi-turnover assays to compare the activities of HACE1 wild type ('FL') with the dimerization-deficient variants toward the physiological substrate RAC1. As HACE1 selectively ubiquitinates the GTP-loaded state of RAC1 (Supplementary Fig. 4a)[27], we used a well-characterized RAC1 variant, Q61L, that constitutively adopts the active conformation[57]. Our studies unveil that HACE1 FL is autoinhibited, while the dimerization-deficient variants efficiently modify RAC1 (Fig. 2c); the extent of the modification can be evaluated from Supplementary Fig. 4b. Monomerization also promotes HACE1 autoubiquitination and free Ub chain formation in the absence of RAC1 (Fig. 2d and Supplementary Fig. 4c). This indicates that the autoinhibition affects the inherent catalytic properties of HACE1, independently of substrate binding. The relative activities of the tested variants correlate with their propensities to form a monomer (ΔN, L11D, L15D>L8D), supporting the notion that dimerization disfavors catalysis. More conservative substitutions of Leu8, Leu11 and Leu15, respectively, with alanine perturb dimerization and catalytic activity less than aspartate but follow the same trend (Extended Data Table 1 and Extended Data Fig. 4a–c). Together, these data confirm that the N-helix is critical for dimerization-induced autoinhibition of HACE1.

Aside from the N-helix, ANK1 contributes intermolecular, polar contacts in the HACE1 dimer. To interrogate their significance, we introduced alanine substitutions of Tyr32, Gln42 and Arg44 (Extended Data Fig. 4d). However, they did not markedly affect the dimerization propensity or the activity of HACE1 (Extended Data Fig. 4e–g), suggesting that ANK1 does not have a major role in dimerization. Nevertheless, the N-terminal portion of ANK1 and adjacent ANKs experience enhanced HDX upon disruption of the dimer (Fig. 1g, Supplementary Fig. 3 and Extended Data Fig. 3). This probably reflects monomerization-induced conformational changes that are communicated from the N-helix to the ANKs.

To evaluate whether the dimerization-induced autoinhibition of HACE1 occurs in a cellular context, we performed co-immunoprecipitations (co-IPs) upon co-expressing HA and GFP (mClover)-tagged HACE1 in HeLa cells (Fig. 2e). Similar to our in-vitro data, HACE1 wild type ('FL')—but not the ΔN, L11D and L15D variants—robustly self-associates in this setting. HACE1 dimerization, observed in vitro, and its self-association in cells thus depend on a common set

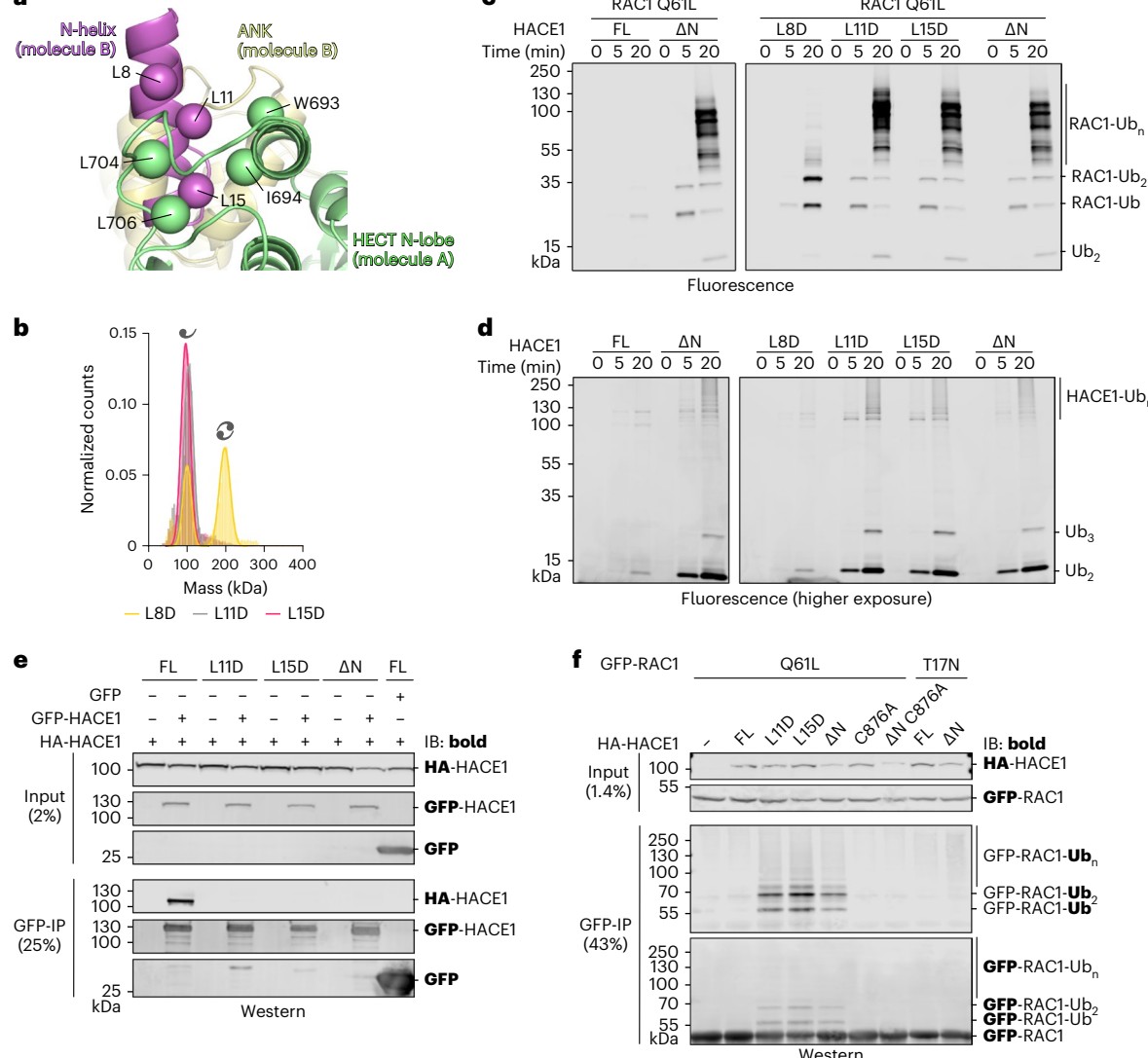

**Fig. 2 | Dimerization of HACE1 confers autoinhibition. a,** Expanded view of the subunit interface of the HACE1 FL cryo-EM structure, highlighting the hydrophobic face of the N-helix and the surrounding network of hydrophobic residues within the small wing of the HECT N-lobe of the adjacent subunit. Owing to the limited resolution of the structure, $C^{\beta}$-atoms (spheres) are displayed instead of full side chains. **b,** MP analysis of full-length HACE1 variants. For MWs and other parameters, see Extended Data Table 1. **c,** Reconstituted multi-turnover ubiquitination assay, monitoring the activity of HACE1 variants toward RAC1 Q61L by SDS–PAGE. Ubiquitinated products are visualized by fluorescence imaging. For the Coomassie-stained gels, see Supplementary Fig. 4b. FL, wild type. **d,** Reconstituted multi-turnover ubiquitination assay, monitoring HACE1

autoubiquitination by SDS–PAGE, analogously to **c.** HACE1 autoubiquitination is less efficient than the ubiquitination of RAC1, necessitating a lower intensity threshold compared to **c** ('higher exposure'). For the Coomassie-stained gels, see Supplementary Fig. 4c. FL, wild type. **e,** Co-IP, monitoring the association of HA-tagged and GFP(mClover)-tagged HACE1 variants in HeLa cell lysates by immunoblotting (IB) against HA and GFP; the monitored antigen is marked in bold. GFP-only serves as a control. FL, wild type. **f,** HeLa cell-based assay, monitoring the ubiquitination of RAC1 variants upon co-transfection of HACE1 variants by IP of RAC1 and IB against Ub and GFP (mClover), respectively (monitored antigen in bold). FL, wild type.

of contacts. Likewise, the activities of transiently transfected HACE1 variants toward RAC1 Q61L negatively correlate with their propensities to oligomerize (Fig. 2f): the dimerization-deficient variants ubiquitinate RAC1 Q61L, whereas no activity is observed for HACE1 FL. This observation holds despite an overall reduced level of HACE1 ΔN, which may be because of enhanced cellular turnover of this variant. Catalytically inactive HACE1 (C876A) and GTP binding-deficient RAC1 (T17N) were used as negative controls. In sum, our results suggest that HACE1 dimerization through hydrophobic contacts of the N-helix confers autoinhibition upon overexpression in cells.

To assess whether HACE1 also self-associates at concentrations near endogenous levels, we used stably transfected HEK293 Flp-In T-REx (Thermo Fisher Scientific) cell lines, expressing GFP (mClover)-tagged

HACE1 FL, L15D and ΔN, respectively, from a tetracycline-inducible promoter. In these cells, without inducer, a small amount of 'leaky' expression occurs that is only slightly higher than the endogenous level (Extended Data Fig. 4h). Co-IPs in this system confirm that HACE1 FL self-associates and that this interaction is sensitive to the L15D substitution or truncation of the N-helix (Extended Data Fig. 4h). N-helix-dependent association of HACE1 FL can thus be detected across a wide concentration range in cell lysates.

### HACE1 phosphorylation may modulate dimerization
We next sought to detect dimerization of endogenous HACE1 in mouse and rat-derived brain tissues, owing to the significance of the ligase in neuronal homeostasis[25,36–39]. HACE1 FL is highly conserved between

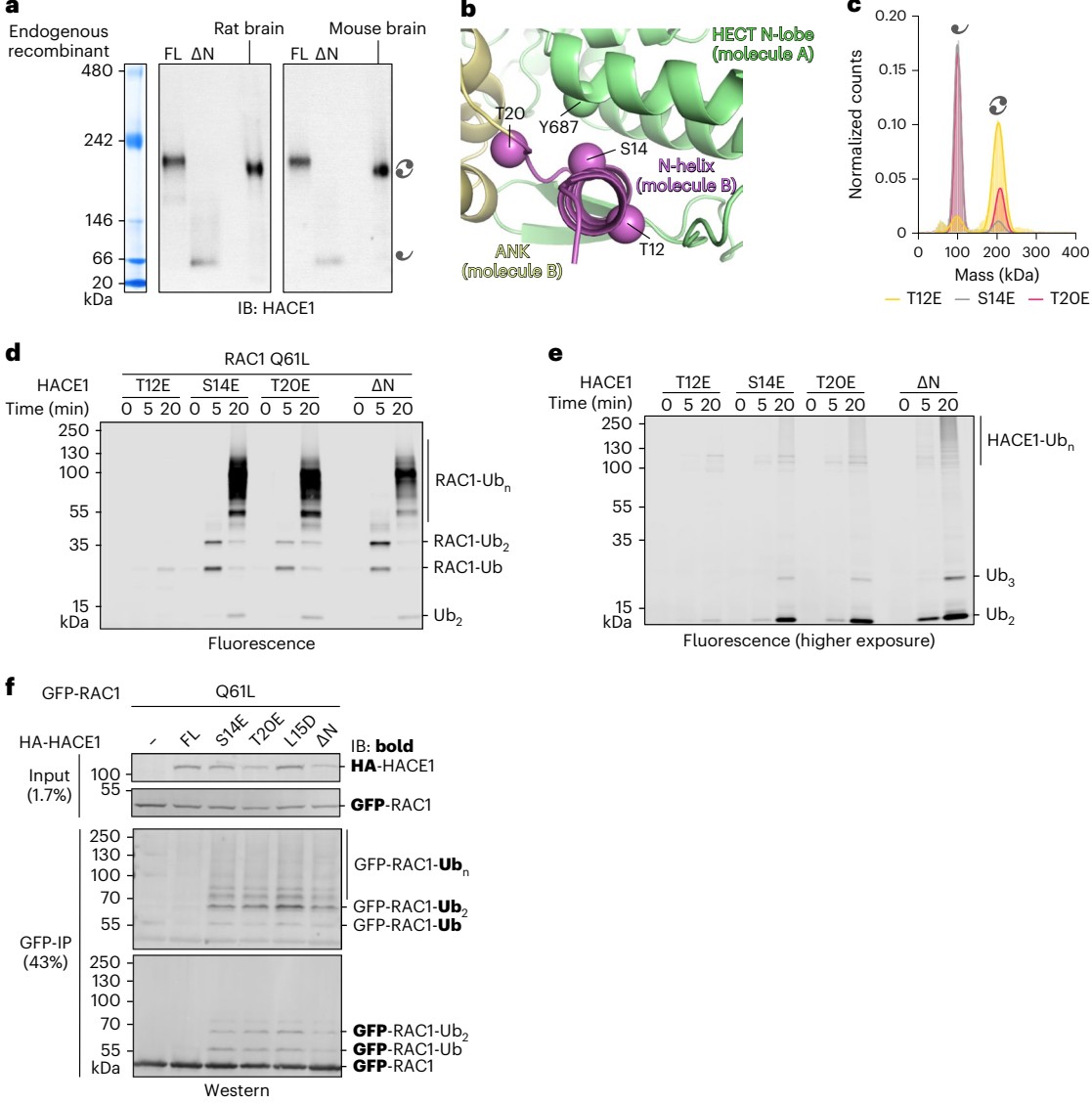

**Fig. 3 | Phospho-mimetic mutations conformationally modulate HACE1 activity. a**, Native PAGE analysis of brain homogenates from mouse and rat compared to recombinant HACE1 FL and ΔN, monitored by IB; Coomassie-stained native marker bands are shown on the left. **b**, Expanded view of the subunit interface of the HACE1 FL cryo-EM structure, highlighting three physiological phosphorylation sites, Ser14, Thr20 and Tyr687, as well as Thr12, which was not found phosphorylated in cells[58]. Owing to the limited resolution of the structure, $C^\beta$-atoms (spheres) are displayed instead of full side chains. **c**, MP analysis of HACE1 variants. For MWs and other parameters, see Extended Data Table 1. **d**, Reconstituted multi-turnover ubiquitination assay, monitoring

the activity of HACE1 variants toward RAC1 Q61L by SDS–PAGE. Ubiquitinated products are visualized by fluorescence imaging. For the Coomassie-stained gel, see Supplementary Fig. 5b. **e**, Reconstituted multi-turnover ubiquitination assay, monitoring HACE1 autoubiquitination by SDS–PAGE, analogously to **d**. HACE1 autoubiquitination is less efficient than the ubiquitination of RAC1, necessitating a lower intensity threshold compared to **d** ('higher exposure'). For the Coomassie-stained gel, see Supplementary Fig. 5c. **f**, HeLa cell-based assays, monitoring the ubiquitination of RAC1 Q61L upon co-transfection of HACE1 variants by IP of RAC1 and IB against Ub and GFP, respectively (monitored antigen in bold). FL, wild type.

human, mouse and rat, reflected by ~97% overall amino acid sequence identity and 100% within the dimerization region (Supplementary Fig. 5a). The brain homogenate was centrifuged at low speed and the supernatant was analyzed by native PAGE with immunoblotting against HACE1. Intriguingly, endogenous HACE1 migrates at ~200 kDa, similar to the recombinant dimer and distinct from the ΔN monomer (Fig. 3a). This suggests that endogenous HACE1 may dimerize and the dimer represents the predominant form in the brain fractions analyzed here. It also raises the important question of which mechanisms regulate HACE1 oligomerization and activity.

To explore such mechanisms, we considered post-translational modifications at the subunit interface. Interestingly, this interface

indeed contains a cluster of three residues—Ser14 and Thr20 of one subunit and Tyr687 of the other (Fig. 3b)—that are physiological phosphorylation sites[58]. Ser14 resides within the critical N-helix, while Thr20 is in the adjacent hinge-loop that connects the N-helix with ANK1. Both residues are tightly embedded at the subunit interface, proximal to Tyr687 in the neighboring HECT N-lobe. We thus speculated that phosphorylation at these sites may modulate HACE1 dimerization. Given that the kinases targeting the sites are not known, we analyzed the consequences of phospho-mimetic substitutions at Ser14 and Thr20. As a control, we generated a phospho-mimetic substitution at Thr12 that has not been found phosphorylated in cells[58] and faces away from the subunit interface. As anticipated, the S14E

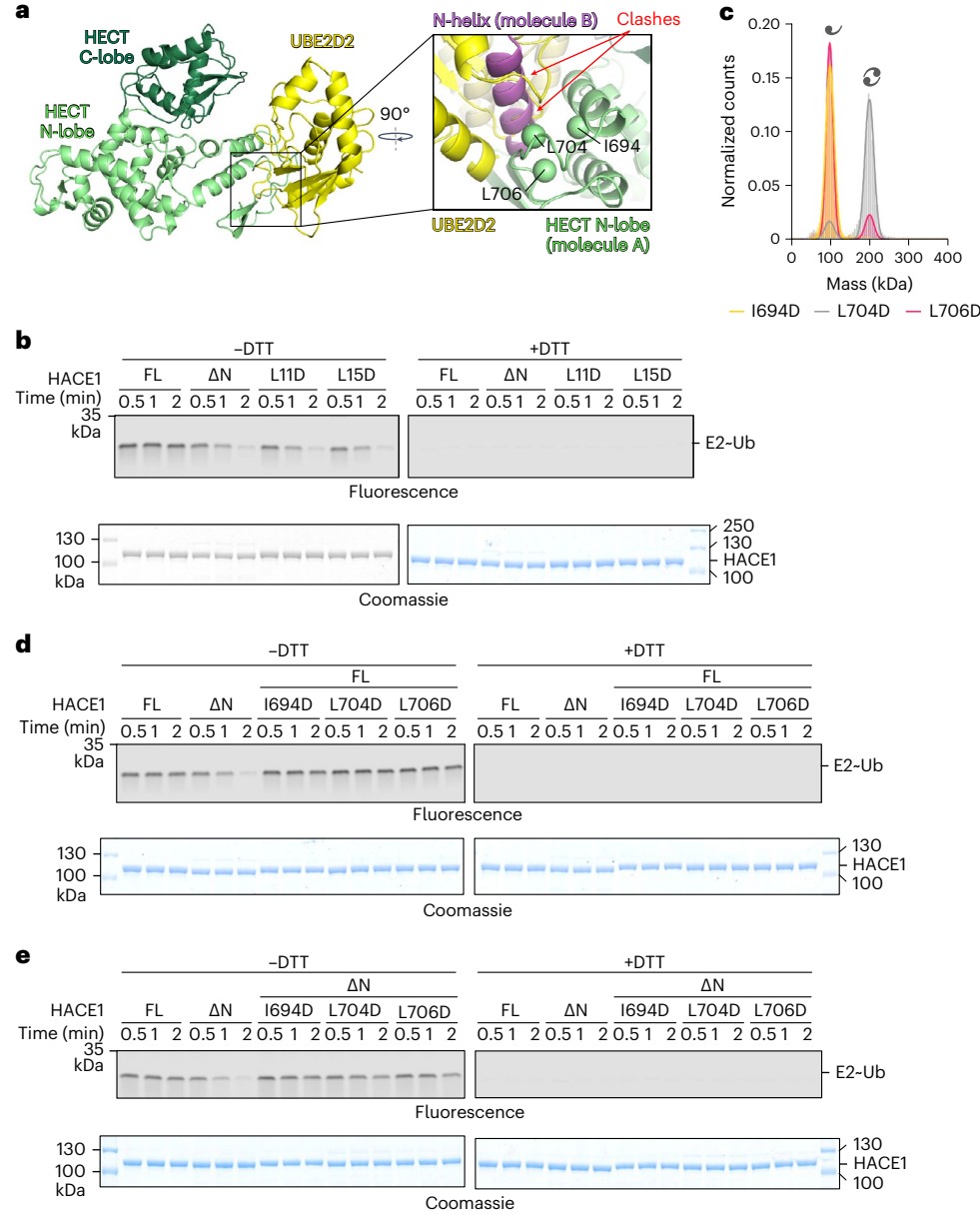

**Fig. 4 | Dimerization inhibits HACE1 at the first catalytic step. a**, Structural model of a complex of the HECT domain of HACE1, extracted from our cryo-EM structure, with UBE2D2, based on a crystal structure of a UBE2D2–NEDD4L HECT domain complex (PDB 3JW0)[16] (left). Expanded view of the model, now in the context of the HACE1 FL dimer, highlighting clashes of UBE2D2 with the N-helix of HACE1 (right). Hydrophobic contact sites of the N-helix in the adjacent N-lobe are shown as $C^\beta$-spheres. **b**, Reconstituted single-turnover assay, comparing the abilities of HACE1 variants to promote Ub discharge from UBE2L3 by SDS–PAGE. The thioester linkage between E2 and Ub is indicated as '~' and sensitive to reducing agent (DTT). E2~Ub is visualized by fluorescence imaging (top); E3 input by Coomassie staining (bottom). For additional gel regions, see Extended Data Fig. 6a. FL, wild type. **c**, MP analysis of HACE1 variants. For MWs and other parameters, see Extended Data Table 1. **d**, Reconstituted single-turnover assay monitoring Ub transfer from UBE2L3 to HACE1 variants by SDS–PAGE, analogously to **b**. For additional gel regions, see Extended Data Fig. 6b. FL, wild type. **e**, Reconstituted single-turnover assay monitoring Ub transfer from UBE2L3 to HACE1 variants by SDS–PAGE, analogously to **b**. For additional gel regions, see Extended Data Fig. 6e. FL, wild type.

and T20E, but not the T12E variant, are predominantly monomeric (Fig. 3c and Extended Data Table 1). Both variants also display enhanced HDX compared to the wild-type dimer ('FL'), with similar profiles to monomeric HACE1 ΔN (Extended Data Fig. 5) and increased substrate and autoubiquitination activities, while the T12E variant is autoinhibited (Fig. 3d,e and Supplementary Fig. 5b,c). Moreover, the S14E and T20E substitutions stimulate HACE1-driven RAC1 ubiquitination in cells (Fig. 3f). These results indicate that phosphorylation of Ser14 and Thr20 may stimulate HACE1 activity by stabilizing its monomeric state, providing a possible mechanism of how the activation of this ligase may be tuned.

## HACE1 dimerization inhibits the first catalytic step

The structure of HACE1 FL provides a rationale for how dimerization inhibits the two-step catalytic cycle: at the subunit interface, the N-helix engages intermolecularly with a region of the HECT N-lobe that coincides with the conserved E2 binding site (Fig. 4a). We thus predicted that dimerization inhibits E3-mediated Ub discharge from the E2. Indeed, single-turnover assays show that HACE1 FL is impaired in Ub discharge compared to the dimerization-deficient variants (Fig. 4b). Interestingly, discharge driven by the activated variants does not cause an equivalent accumulation of thioester-linked 'E3~Ub' (Extended Data Fig. 6a). This implies that the thioester between active HACE1 and Ub

is highly reactive, enabling rapid Ub discharge even without substrate, analogous to other HECTs[50,59].

We next dissected the role of hydrophobic residues of the N-lobe (Ile694, Leu704 and Leu706) that intermolecularly engage with the N-helix (Figs. 2a and 4a). The I694D and L706D substitutions render HACE1 monomeric, whereas the L704D variant remains predominantly dimeric (Fig. 4c and Extended Data Table 1). This can be rationalized by Ile694 and Leu706 contacting the critical Leu15 in the N-helix, whereas Leu704 is located within a loop that may accommodate an aspartate substitution without disrupting the dimer. As expected, neither of the variants promotes Ub discharge from the E2, HACE1 autoubiquitination or RAC1 ubiquitination, either in the HACE1 FL or the ΔN context (Fig. 4d,e and Extended Data Fig. 6b–f). This global activity defect can be attributed to the mutated residues residing within the E2 binding site (homologous residues, Ile743, Leu751 and Leu753 of NEDD4L[16]; Leu642, Met653 and Ile655 of UBE3A[60]).

We thus suggest that HACE1 dimerization inhibits productive interactions with the E2, thereby efficiently shutting down activity at the first catalytic step. Interestingly, the dimerization is also incompatible with the requirements of the subsequent reaction steps. For example, Ub chain elongation by HECTs often requires the engagement of a regulatory Ub at an 'exosite' of the HECT N-lobe[61–64]. Although this site, including a key phenylalanine (Phe715), is conserved in HACE1, it is occluded in the dimer (Extended Data Fig. 6g). Moreover, we show below that HACE1 dimerization prohibits substrate recognition.

### Selective crosslinking captures a HACE1–substrate complex

Having established that HACE1 activity requires monomerization, we set out to determine a structure of monomeric HACE1 ΔN with the substrate RAC1. Consistent with E3-substrate interactions being transient, HACE1 and RAC1 do not co-elute during SEC, regardless of the activation state of either protein (Extended Data Fig. 7a,b). Yet slight broadening of the elution peaks is observed specifically upon mixing HACE1 ΔN with RAC1 Q61L, hinting at a selective interaction. We thus used a mechanism-based crosslinking strategy based on the short (1.5 Å) amino and sulfhydryl-reactive succinimidyl iodoacetate (SIA)[65] to trap the crucial state in which a lysine ubiquitination site of RAC1 is juxtaposed to the catalytic Cys876 of HACE1 (Fig. 5a). As HACE1 predominantly ubiquitinates Lys147 of RAC1 (Extended Data Fig. 7c), we expected the crosslinking to confer selectivity. Indeed, a small amount of a ~124 kDa HACE1–RAC1 complex forms only when both Cys876 of HACE1 and Lys147 of RAC1 are present (Fig. 5b; 'XL-1'). MS analyses of this complex confirm that SIA crosslinks specifically these two residues; no additional intermolecular crosslinks between HACE1 ΔN and RAC1 Q61L were detected (Fig. 5c and Supplementary Fig. 6a). This demonstrates that HACE1 transiently orients its substrate such that it is primed for Ub transfer, even without other reaction components. Consistent with the flexibility of the HECT domain and the transient nature of the HACE1–RAC1 interaction, the crosslinking efficiency appears to be limited by an alternative crosslink between Cys876 and Lys689 of HACE1 that is juxtaposed in the inverted-T conformation (Fig. 5c and Supplementary Fig. 6b) and by higher-order crosslinks between Lys396 and Cys441 within the flexible loop 2 of HACE1 (Fig. 5c and Supplementary Fig. 6c). However, the unique HACE1–RAC1 crosslink allowed us to purify the desired 1:1 complex (XL-1), validate its size by mass photometry (Fig. 5d and Extended Data Table 1) and determine its structure.

### Cryo-EM structure explains HACE1 selectivity for GTP-RAC1

Our cryo-EM structure of the HACE1 ΔN–RAC1 Q61L complex has a resolution of 4.2 Å (Table 1, Fig. 5e,f and Extended Data Fig. 8a–e), with the map being best defined for the ANKs, the MID and the large wing of the HECT N-lobe of HACE1 as well as the GTPase core of RAC1. Model building was facilitated by an AF2 starting model (Supplementary Fig. 7). In the complex, the concave platform of HACE1 adopts a similar shape as in the dimer. However, the HECT domain is drastically

rearranged relative to the ANKs, with changes also affecting the interjacent MID (Extended Data Fig. 8f). The crosslinked HECT domain now adopts an L-conformation, in which the active site is juxtaposed to Lys147 of RAC1. Transitioning of the C-lobe between the inverted-T and the L-conformation (here trapped by SIA) can thus occur, in principle, without a 'donor' Ub attached to the E3. Engagement of the donor, however, stabilizes the C-lobe in the L-state, lending directionality to the catalytic cycle[18,49,50,66].

The structure shows RAC1 cradled by the concave face of the ligase platform, contacting the majority of the ANKs (Fig. 5f). The binding mode is incompatible with the HACE1 dimer, highlighting another layer of autoinhibition (Extended Data Fig. 8g). Intriguingly, the HACE1 binding site of RAC1 comprises the critical switch-I and switch-II regions and the interjacent β2-strand that are known to undergo nucleotide-dependent rearrangements (Fig. 5g)[44]. HACE1 recognizes the switch regions in an ordered conformation, as imposed by the binding of GTP. By contrast, the switch regions are dynamic when bound to GDP, which presumably disfavors interactions with HACE1. The ligase thus conformationally discriminates between the nucleotide-loading states of RAC1 and selects the GTP-bound form as a substrate.

### The RAC1 binding mode of HACE1 is specific

To interrogate the HACE1–RAC1 complex without crosslinking in solution, we compared the HDX profiles of HACE1 ΔN and RAC1 Q61L, when mixed, to those in isolation (Fig. 5h and Supplementary Figs. 8 and 9). Although the changes detected for either protein are rather small, probably because of the transiency of the interaction, they recapitulate key features of the cryo-EM structure: in the presence of HACE1, RAC1 experiences HDX reductions particularly in the switch-II region and in the β3-strand flanking switch-I. Peptides covering switch-I were detected for *apo* RAC1 but are missing in the presence of HACE1. This may reflect a locally reduced proteolytic cleavage efficiency of RAC1 when bound to HACE1, consistent with the observed binding mode. Elevated HDX of HACE1-bound RAC1 occurs in α-helix 5, close to the critical β-sheet, highlighting propagated, binding-induced perturbations in the GTPase. HACE1 shows reduced HDX in all ANKs, albeit to various degrees. Elevated HDX is detected in the hinges between the MID and HECT domain and between the wings of the HECT N-lobe, revealing substrate-binding-induced allosteric changes.

To interrogate whether the N-helix contributes to RAC1 binding by monomerized HACE1, we analyzed HDX in the dimerization-deficient, full-length S14E and I694D variants, with and without RAC1 Q61L (Extended Data Fig. 9). Both variants show RAC1-induced HDX reductions in the ANKs, as seen for HACE1 ΔN, indicative of a common binding mode. In addition, RAC1 enhances HDX near the N-terminus and in the N-lobe of HACE1 S14E, recapitulating changes seen upon disruption of the dimer (Supplementary Fig. 3 and Extended Data Fig. 3). This suggests that RAC1 binding may stabilize the monomeric state of this variant. By contrast, HACE1 I694D does not experience RAC1-induced HDX enhancements in these regions, suggesting that it is fully monomeric. Instead, elevated HDX is observed in other areas of the N-lobe, probably reflecting mutation-induced changes in HECT domain dynamics. Importantly, neither the S14E variant nor the I694D variant displays HDX changes indicative of an interaction between the N-helix and RAC1. This suggests that the binding mode identified structurally and supported by common RAC1-induced HDX reductions in all tested dimerization-deficient variants is independent of the N-helix.

To evaluate the functional significance of the HACE1–RAC1 binding mode, we mutationally analyzed three contact zones (Fig. 6a,b and Supplementary Fig. 10a) with regard to RAC1, E3 (Fig. 6c–i) and OPTN ubiquitination (Fig. 6j). Zone I involves ANK 5 of HACE1, with Gln173 and Asn174 forming polar interactions with Asn26 and Thr24 of RAC1. Alanine substitutions of Gln173, Asn174 and Asn26, alone or in combination, reduce RAC1 ubiquitination (Fig. 6c,d and Supplementary Fig. 10b,c).

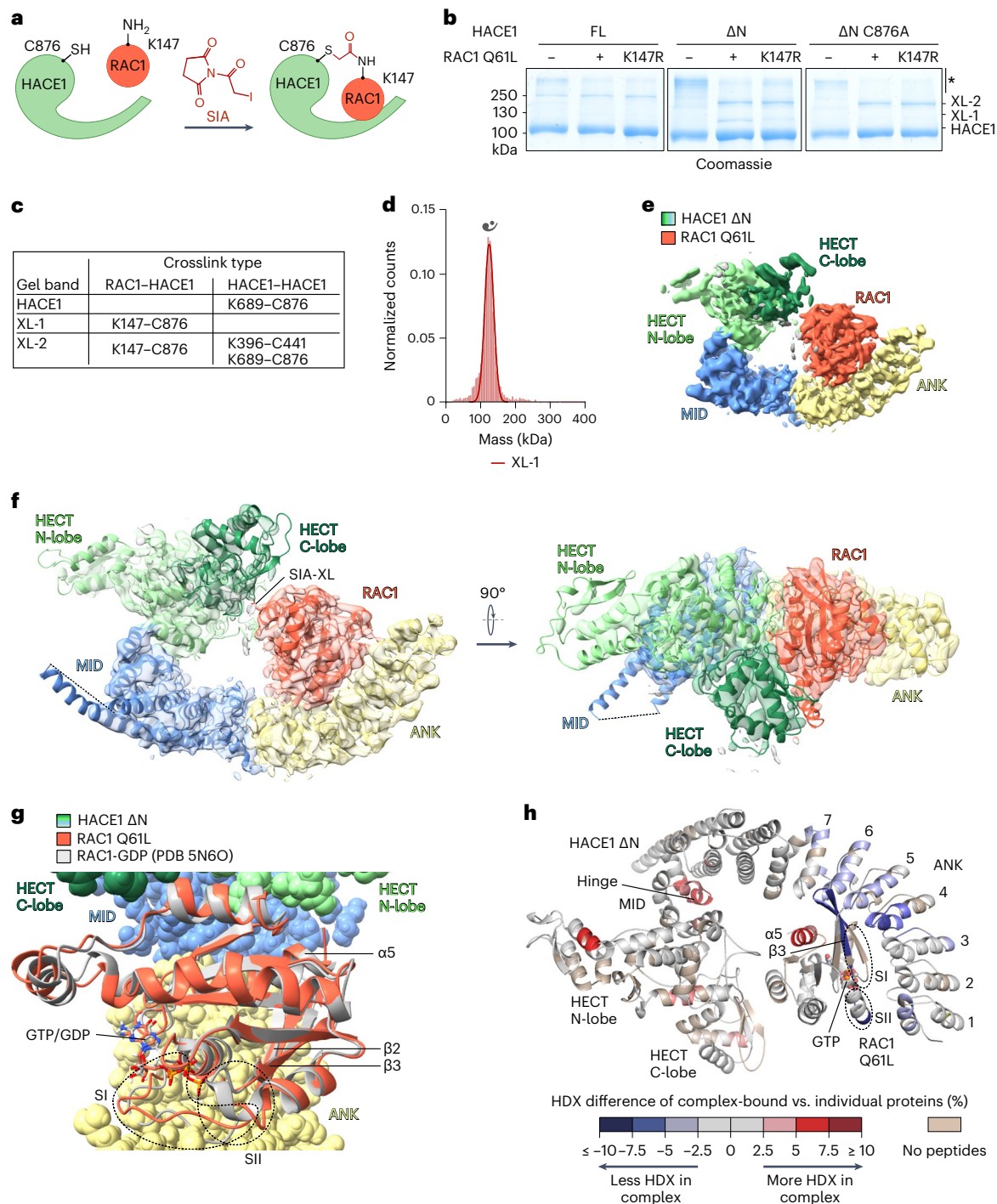

**Fig. 5 | Reconstitution and structure determination of a HACE1–RAC1 complex. a**, Cartoon illustrating the crosslinking (XL) of the catalytic Cys876 of HACE1 to the Lys147-ubiquitination site of RAC1 with the heterobifunctional SIA. **b**, SDS–PAGE-based selectivity analysis of SIA XL, using HACE1 and RAC1 variants, visualized by Coomassie staining. Three distinct bands (HACE1, XL-1, XL-2) were analyzed by MS in **c**. Higher-order species are marked by '*'. Note that XL-1 requires both Lys147 of RAC1 and Cys876 of HACE1. **c**, MS analysis of crosslinks from **b**. For mass spectra, see Supplementary Fig. 6. **d**, MP analysis of the purified, crosslinked HACE1 ΔN–RAC1 Q61L complex (XL-1). For MW and other parameters, see Extended Data Table 1. The symbol denotes a 1:1 complex. **e**, Cryo-EM map of the HACE1 ΔN–RAC1 Q61L complex with individual domains colored. **f**, Composite cryo-EM map and structure of the complex in two orientations.

Where loops are missing, secondary structure elements are connected by a dashed line. The SIA-based crosslink is marked. Domains are colored as in **e**. **g**, Expanded view of a superposition of the structure of the HACE1–RAC1 complex (HACE1 as surface; RAC1 Q61L as cartoon), with a crystal structure of GDP-bound RAC1 (PDB 5N6O)[79]. Relevant elements of RAC1 are labeled (SI, switch-I; SII, switch-II), following RHO-GTPase nomenclature. **h**, HDX differences between the HACE1 ΔN–RAC1 Q61L complex and either protein in isolation, mapped onto the cryo-EM structure. The colors reflect changes upon complex formation. Regions not covered by the proteolytic digestion are shown in beige. The orientation of the structure is flipped compared to **f**. For HDX data, see Supplementary Figs. 8 and 9 and Supplementary Data 2.

In zone II, Thr24 of RAC1 and Val140 in ANK 4 were substituted by leucine to impose steric restraints (Fig. 6e and Supplementary Fig. 10d). Although V140L reduces HACE1 activity toward RAC1 (as previously reported[67]), T24L enhances it. Zone II is thus exquisitely sensitive to perturbations with activating or inhibiting consequences. Disrupting contacts in zone III (Arg107 and Asn108 in ANK 3 of HACE1 and Asp38 at the base of the β2-strand of RAC1) generally interferes with RAC1 ubiquitination (Fig. 6f,g and Supplementary Fig. 10e,f). The tested *HACE1* mutations in all three zones also perturb ligase activity toward active RAC1 upon overexpression in HeLa cells (Fig. 6i). R107A, Q173A and N174A appear to be more disruptive than N108A and V140L, recapitulating the trend observed in vitro.

Importantly, the same mutations do not cause a loss of HACE1 activity in RAC1-independent autoubiquitination and free Ub chain formation (Fig. 6h and Supplementary Fig. 10g). One variant (R107A) appears even slightly more active in this context (see also Fig. 6j and Supplementary Fig. 10h). Alterations in the identified RAC1 binding site thus do not inhibit HACE1 per se but specifically affect its activity toward RAC1. Consistently, the tested HACE1 variants support the ubiquitination of an alternative substrate, OPTN (Fig. 6j and Supplementary Fig. 10h), implying that HACE1 recognizes RAC1 and OPTN in distinct ways. Finally, a comparison of the activities of HACE1 FL and ΔN toward OPTN confirms that the HACE1 dimer is autoinhibited in a substrate-independent manner, and release of the N-helix generally promotes activity. Together, these findings illustrate how global and specific parameters converge in regulating the catalytic activity and substrate selectivity of HACE1.

## Discussion

Ub ligases are key determinants of the specificity and spatiotemporal control of ubiquitination. We discovered that the HECT HACE1 is conformationally regulated through dimerization (Fig. 7a). The yin–yang-like, autoinhibited structure we determined resembles an independently characterized one that was recently reported[51]. The HACE1 dimer interface occludes the E2 binding site of the HECT N-lobe, thus blocking Ub transfer from the E2. Moreover, the dimer is incompatible with subsequent reaction steps, including RAC1 recruitment. Although structurally distinct, analogous autoinhibition mechanisms affecting the first catalytic step efficiently restrict the activities of E2s[68–70] and RBR-type ligases[71]. Autoinhibition is, therefore, a widespread theme across different classes of ubiquitination enzymes, highlighting the vital cellular requirement of ubiquitination to be stringently regulated.

Purified HACE1 FL is predominantly dimeric in vitro, with no monomers detected. Similar to other work[51], however, we observe monomer-like particles on cryo-EM grids, which we attribute to dimers being damaged during the freezing. We show that overexpressed HACE1 self-associates through the N-helix and is firmly autoinhibited in cells; dimerization is also seen at endogenous protein levels in animal-derived brain homogenate fractions. This raises the question of which factors modulate HACE1 oligomerization and activity. We propose that the phosphorylation of linchpin sites in the N-terminal region[58] may stabilize the monomeric, active state. The context and extent of such phosphorylation and the kinase(s) responsible, however, remain to be identified. To this end interactomic studies may provide entry points[72]. It is also conceivable that HACE1 is regulated by yet unknown interactors of the N-helix or recruitment to membranes. For example, OPTN was reported to activate HACE1 toward RAC1 at the plasma membrane[28] and HACE1 selectively interacts with RAB-family GTPases, including RAB1, RAB4 and RAB11 (refs. 32,73), which may program its localization, substrate exposure or activity. As for RAC1, HACE1 interacts with RABs in a GTP-loading-dependent manner. RAB11 is ubiquitinated by HACE1 at a site homologous to Lys147 of RAC1 (ref. 73), suggesting that the ligase may allow for certain redundancy in the recognition of small GTPase substrates, at least upon overexpression. Finally, HACE1 may be regulated at the transcriptional level: public databases[74] contain several human HACE1 isoforms, of which two lack 34 N-terminal residues (NCBI NP_001308012.1 and NP_001337483.1), including the critical N-helix. It will be interesting to explore whether these isoforms are expressed at physiologically relevant levels and which distinct activities and specificities they may confer.

The autoinhibition mechanism that we uncovered provided the basis for reconstituting a complex of an active HACE1 monomer with RAC1. To stabilize this complex, we used mechanism-based crosslinking, exploiting the native proximity of the substrate's lysine ubiquitination site to the catalytic cysteine of the ligase[65]. The strategy introduces only a short spacer and confers selectivity without unnatural amino acid handles, which can be challenging to incorporate. While the selectivity of this crosslinking approach is system-dependent, it may be applicable to structural analyses of other complexes of catalytic cysteine-dependent E3s with substrates.

We demonstrate that the introduced lysine–cysteine crosslink captures RAC1 in a conformation poised for ubiquitination and the HECT domain in the relevant L-conformation for Ub discharge to the substrate. Notably, this arrangement is compatible with the conserved binding mode of the donor Ub to the HECT C-lobe (Fig. 7b)[16–18,49,50,66,75]. The HACE1 binding site of RAC1 shows the critical switch regions in an ordered conformation, as induced by GTP, providing a rationale for why HACE1 selectively modifies the active state of RAC1. Numerous cancer patient-derived mutations in *HACE1* (Gln173 (analyzed here; Fig. 6c), Ile71, Ile132, Arg143, Asp161, Gly175 and Arg547 (ref. 76)) coincide with the RAC1 binding site and may alter its properties. Expression of one of them, G175S, was shown to suppress RAC1 ubiquitination and promote anchorage-independent cell growth[67]. The in-vivo impact of the mutations on tumorigenesis, however, awaits future investigation. It will also be crucial to dissect how HACE1 encodes specificity for structurally diverse substrates beyond RAC1. While the recognition of OPTN also requires the ANKs[30], our studies indicate a binding mode distinct from RAC1.

Aside from HACE1, only two full-length human HECTs (HUWE1 (ref. 45) and UBR5 (refs. 10,11,47,48,50)), *Nematocida* Huwe1 (ref. 46) and yeast Ufd4 (ref. 49) have been visualized by cryo-EM. Our results thus considerably contribute to defining architectural paradigms in

**Fig. 6 | Functional analysis of the HACE1–RAC1 interface. a**, Expanded view of the cryo-EM structure of the HACE1 ΔN–RAC1 Q61L complex from Fig. 5f, highlighting three zones. The ANKs are numbered. **b**, Detailed views of zones I–III from **a**, with relevant backbone and side-chain contacts highlighted as sticks. The resolution of the structure allowed for placement of side chains in zones I and II with some confidence (for the map, see Supplementary Fig. 10a). **c,d**, Mutational analysis of zone I, using reconstituted multi-turnover ubiquitination assays, monitoring the activity of HACE1 ΔN variants toward RAC1 Q61L variants. Ubiquitinated products are visualized by fluorescence imaging. For the Coomassie-stained gels, see Supplementary Fig. 10b,c. **e**, Mutational analysis of zone II, using reconstituted multi-turnover ubiquitination assays, analogous to **c** and **d**. For the Coomassie-stained gels, see Supplementary Fig. 10d. **f,g**, Mutational analysis of zone III, using reconstituted multi-turnover ubiquitination assays, analogous to **c** and **d**. For the Coomassie-stained gels, see Supplementary Fig. 10e,f. **h**, Reconstituted multi-turnover ubiquitination assay, monitoring HACE1 autoubiquitination. Ubiquitinated products are visualized by fluorescence imaging. The activity is less efficient than RAC1 ubiquitination, necessitating a lower intensity threshold compared to **c** and **d** ('higher exposure'). For the Coomassie-stained gels, see Supplementary Fig. 10g. **i**, HeLa cell-based assay, monitoring the ubiquitination of RAC1 Q61L upon co-transfection of HACE1 variants, IP of RAC1 and IB against Ub and GFP, respectively (monitored antigen in bold). FL, wild type. **j**, Reconstituted multi-turnover ubiquitination assay, monitoring the activity of HACE1 ΔN variants toward OPTN. Ubiquitinated products are visualized by fluorescence imaging. The activity is less efficient than RAC1 ubiquitination, necessitating a lower intensity threshold compared to **c** and **d** ('higher exposure'). For the Coomassie-stained gel, see Supplementary Fig. 10h. FL, wild type.

this ligase family. (1) Interestingly, the dimensions and ring shape of dimeric HACE1 resemble the structural core of monomeric HUWE1, which forms a flexible solenoid (Supplementary Fig. 11). This similarity arises from the α-helical platform of HACE1 having similar length and curvature to the two armadillo-repeat regions that flank the HECT and tower domains of HUWE1. Helical-repeat platforms also recur in

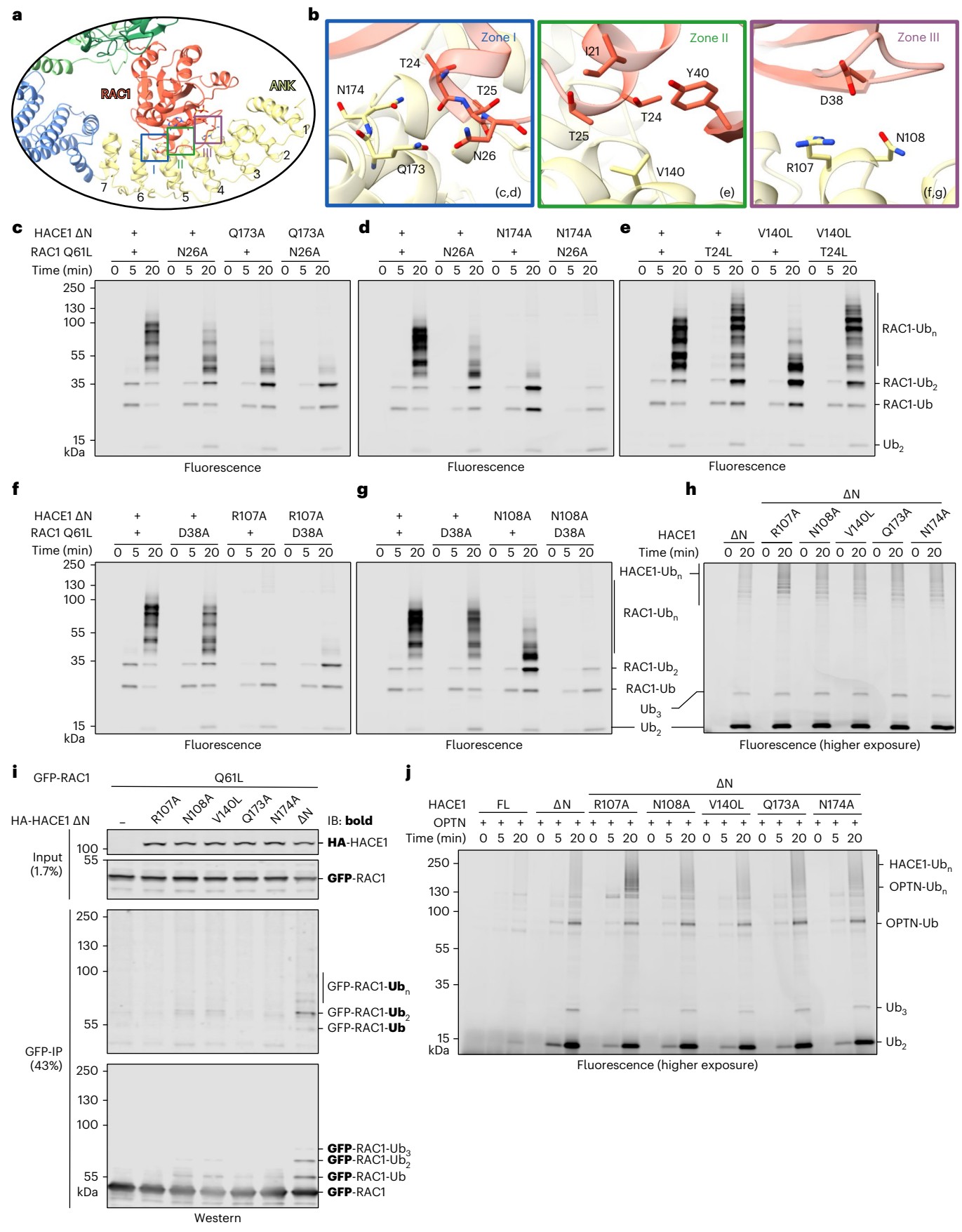

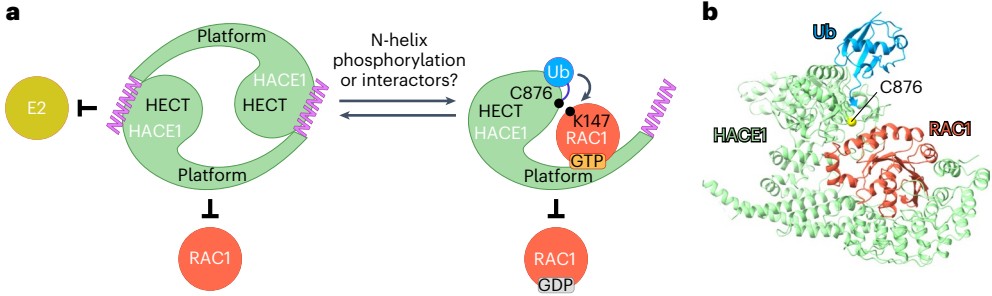

**Fig. 7 | Mechanisms of autoinhibition and substrate recognition by HACE1.**
**a**, Cartoon summarizing key insights from this study: HACE1 is autoinhibited by dimerization, with the N-helix making critical intermolecular contacts with the HECT domain at the subunit interface. The yin–yang-like dimer prohibits Ub transfer from the E2 and is incompatible with the engagement of RAC1 as a substrate. HACE1 activation requires release of the N-helix, allowing for productive interactions with the E2 and substrates, such as RAC1. The RAC1

binding mode explains how HACE1 selects for GTP-loaded over GDP-loaded RAC1. We propose that phosphorylation of linchpin sites within the N-terminal region of HACE1 may stabilize the monomeric, active state. Additionally, interaction partners may regulate the conformation of the N-helix and thus HACE1 activity. **b**, Donor Ub modeled into the cryo-EM structure of the HACE1 ΔN–RAC1 Q61L complex, based on a superposition of the HECT domain with PDB 6XZ1 (ref. 66).

the structures of UBR5 (refs. 10,11,47,48,50), Ufd4 (ref. 49) and AF2 predictions of additional human HECTs, such as HECTD1. Although it is unclear which precise functional requirements underlie the evolution of this architectural core, we speculate that the overall mobility and internal dynamics of the conserved HECT domain impose restraints on the geometry of the associated substrate-presenting platform. Notably, the conformational cycle of the HECT domain during its interactions with E2, donor Ub and substrates, as established by pioneering work on HECT ligase fragments[16–18,66,77], appears to be conserved within the Ub transfer complexes of the full-length enzymes[49,50]. (2) In all structurally resolved HECTs, the platform is decorated with flexible insertions conferring specificity or regulation. For example, a regulatory phosphorylation site resides in loop 2 of HACE1 (ref. 78) and substrate or Ub-binding motifs are inserted into the platforms of HUWE1 (ref. 45,46) and UBR5 (refs. 10,11,47,48,50). The dynamic nature of these regions may allow HECTs to recruit diverse substrates for modification within the constraints of an overall similar catalytic platform. (3) In addition to such local flexibility, global plasticity has emerged as a regulatory paradigm in HECTs. In HACE1, this manifests as the ability to limit activity through dimerization. UBR5 adopts oligomeric forms, which contribute to substrate recognition[10,11] and the positioning of Ub during chain formation[50]. Finally, the HUWE1 solenoid appears to be capable of opening up, highlighting considerable global plasticity[45,46]. To further explore these concepts and distill mechanistic idiosyncrasies of HECTs requires snapshots of additional substrate-bound complexes, along with insights into their directional transitions.

## Online content

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

[1]Research Group 'Ubiquitin Signaling Specificity', Max Planck Institute for Multidisciplinary Sciences, Göttingen, Germany. [2]Research Group 'Bioanalytical Mass Spectrometry', Max Planck Institute for Multidisciplinary Sciences, Göttingen, Germany. [3]Department of Molecular Biology, University Medical Center Göttingen, Göttingen, Germany. [4]'Bioanalytics', Department of Clinical Chemistry, University Medical Center Göttingen, Göttingen, Germany. [5]'Multiscale Bioimaging: from Molecular Machines to Networks of Excitable Cells', University of Göttingen, Göttingen, Germany. [6]Department of Chemistry, Philipps University Marburg, Marburg, Germany. [7]Center for Synthetic Microbiology, Philipps University Marburg, Marburg, Germany. [8]Department of Molecular Biology, Max Planck Institute for Multidisciplinary Sciences, Göttingen, Germany. [9]These authors contributed equally: Jonas Düring, Madita Wolter. ✉e-mail: sonja.lorenz@mpinat.mpg.de

# Methods

## DNA constructs

Plasmids encoding human *HACE1* wild type (pGEX-6P-1) and C876A (pCDNA3.1) were provided by Yanzhuang Wang (University of Michigan, MI, USA)[41]. For bacterial expression, the genes were cloned into pSKB2 (ref. 80), encoding an N-terminal, 3C protease-cleavable His$_6$-tag, and pN-HZ10-ZZ(3) (provided by Dirk Görlich, Max Planck Institute for Multidisciplinary Sciences (MPI NAT), Göttingen, Germany), encoding an N-terminal TEV protease-cleavable His$_{10}$-tandem Z (IgG-binding domain of protein A) tag. For expression in Sf9 cells, *HACE1* was cloned into a pFastBac derivative (provided by Patrick Cramer, MPI NAT), encoding an N-terminal TEV protease-cleavable His$_6$-MBP tag. For expression in mammalian cells, *HACE1* was sub-cloned from pSKB2 into pcDNA3.1 (Thermo Fisher Scientific), retaining the N-terminal, 3C protease-cleavable His$_6$-tag. For mammalian cell-based IPs and ubiquitination assays, *HACE1* was cloned into pCMV-mClover-GW (provided by Melina Schuh, MPI NAT), and the 3C protease-cleavable His$_6$-tag encoded by pcDNA3.1-HACE1 was replaced with a HA-tag. For the generation of stably transfected HEK293 Flp-In-T-REx cell lines, *mClover* from pCMV-mClover-GW and the *HACE1* constructs from pcDNA3.1 were sub-cloned into pcDNA5.0 FRT TO FRT/TO (Thermo Fisher Scientific).

For bacterial expression of RAC1, a codon-optimized gene (Integrated DNA Technologies) was inserted into pSKB2. For mammalian cell-based experiments, we used pcDNA3-EGFP-RAC1(T17N) and pcDNA3-EGFP-RAC1(Q61L), provided by Klaus Hahn (Addgene nos. 13721 and 13720)[81].

The *OPTN* gene in pDEST17-hOPTN, provided by Jon Ashwell (Addgene no. 23053)[82], was sub-cloned into pSKB2. Cloning and mutagenesis were performed with restriction-free methods. Oligonucleotide sequences are provided in Supplementary Data 4.

## Protein preparation

Ub[83], fluorescently labeled Ub (IRDye 800CW maleimide; LI-COR)[70], UBA1 (ref. 83) and UBE2L3 (ref. 66) were prepared as described. Other bacteria-based preparations generally included expression in *E. coli* BL21(DE3), induction with 0.5 mM IPTG, immobilized nickel ion-based affinity chromatography (IMAC), proteolytic tag removal (unless indicated otherwise) and SEC.

HACE1 variants were expressed at 18 °C overnight and cells were lysed in 50 mM HEPES (pH 8.0), 200 mM NaCl, 20 mM imidazole, 5 mM β-mercaptoethanol (β-ME), containing protease inhibitors (Roche). The same buffer was used for IMAC, with an additional 300 mM imidazole and no protease inhibitors. Protease cleavage was performed during overnight dialysis into in 50 mM HEPES (pH 8.0), 150 mM NaCl and 1 mM β-ME at 4 °C, followed by IMAC in the same buffer and SEC in 50 mM HEPES (pH 8), 150 mM NaCl and 5 mM DTT (HiLoad Superdex 16/600 200 pg; Cytiva). HACE1 expression in Sf9 and HEK293F cells is described in the Supplementary Methods; the proteins were purified in the same way.

RAC1 variants were expressed at 30 °C for 4 h and cells were lysed in 30 mM HEPES (pH 8.0), 150 mM NaCl, 1 mM MgCl$_2$, 20 mM imidazole and 5 mM β-ME. The same buffer was used for IMAC, with an additional 300 mM imidazole during elution. If applicable, protease cleavage was performed at 4 °C during overnight dialysis into 30 mM HEPES (pH 8.0), 150 mM NaCl and 1 mM β-ME. SEC was performed in 30 mM HEPES (pH 8.0), 50 mM NaCl, 5 mM MgCl$_2$ and 3 mM DTT (Superdex 16/600 75 pg; Cytiva). Nucleotide exchange for wild-type RAC1 followed published procedures[32]; excess nucleotide was removed by desalting (HiTrap; Cytiva).

OPTN was expressed at 30 °C overnight. Cells were lysed in 30 mM HEPES (pH 8.0), 500 mM NaCl, 20 mM imidazole and 5 mM β-ME. The same buffer was used for IMAC, with an additional 300 mM imidazole for elution. SEC was performed in 50 mM HEPES (pH 8.0), 150 mM NaCl and 2 mM DTT (Superdex 16/600 200 pg; Cytiva).

## Analytical SEC

Analyses were performed of 20 µM (HACE1), 80 µM (RAC1) or mixtures thereof (molar ratio, 1:4) in 50 mM HEPES (pH 8.0), 150 mM NaCl and 1 mM DTT (Superdex 200 Increase 3.2/300; Cytiva), using an ÄKTA Micro (Cytiva) at 4 °C.

## Mass photometry

HACE1 samples were measured at 20–40 nM concentration in 25 mM HEPES (pH 8.0) and 150 mM NaCl with a One$^{MP}$ mass photometer (Refeyn). A calibration curve was generated with BSA. The 60 s movies were acquired for a medium-acquisition area with AcquireMP (Refeyn, v.1.1.3) and analyzed with DiscoverMP v.2023_R2 (Refeyn); Gaussian fits were generated with PhotoMol[84] and graphs with Prism 9 (GraphPad).

## HACE1–RAC1 crosslinking and MS

For analytical crosslinking, 1 µM HACE1 and 5 µM His$_6$-tagged RAC1 were incubated in PBS at 4 °C for 15 min. Then, 30 µM SIA (in DMSO) was added (0.2% v/v DMSO final), following shaking at room temperature (20–22 °C) for 30 min, quenching with SDS-loading dye and SDS–PAGE. For MS, HACE1 ΔN and RAC1 Q61L were treated as above but crosslinked for 1 h. Following in-gel chymotryptic digestion, peptides were analyzed with an Exploris 480 (Thermo Fisher) coupled to a Dionex UltiMate 3000 uHPLC with a homemade 30 cm C18 column. Crosslinks were identified with pLink v.2.3.11 (ref. 85). SIA linker and mono-mass were set to 39.995 and 58.005 Da, respectively. Only crosslinks of protein N-termini or lysines to cysteines were considered. The underlying custom database encompassed proteins identified by MaxQuant v.2.1.4.0 in the same data set (including contaminants). For preparative applications, crosslinking mixtures were prepared as above but were incubated for 60 min and quenched by 60 min incubation with 20 mM β-ME, 20 mM imidazole and 200 mM Tris (pH 7.6) on ice. The crosslinked complex was purified via the His$_6$-tag on RAC1 by IMAC using 50 mM HEPES (pH 8.0), 20 mM imidazole, 5 mM β-ME and 50 mM NaCl for binding and the same buffer including 400 mM NaCl for washing. Elution was performed with a 0–300 mM imidazole gradient in the same buffer containing 100 mM NaCl, followed by SEC (Superdex Increase 3.2/300 200 pg; Cytiva) in 20 mM HEPES (pH 8.0), 50 mM NaCl and 3 mM DTT.

## Cryo-EM

**HACE1 FL.** The SEC peak fraction of HACE1 was diluted to 0.7 mg ml$^{-1}$ and 3 µl was applied to freshly glow-discharged R2/1 Cu400 grids (Quantifoil). The grids were blotted at 4 °C and 95% humidity, using a Mark IV Vitrobot (Thermo Fisher) (blotting force, 3; 7 s) and plunged into liquid ethane. Data were collected with SerialEM[86] in counting mode on a 300-keV Titan Krios transmission electron microscope (Thermo Fisher Scientific) with a Gatan Quantum LS energy filter (slit width, 20 eV) and a K3 direct electron detector (pixel size, 0.834 Å; exposure, 1 e$^-$ per Å$^2$ per frame; overall electron dose per image, 40 e$^-$ Å$^{-2}$). Motion correction, dose weighting, contrast-transfer function estimation and particle picking were accomplished using Warp v.1.0.9 (ref. 87). A total of 3.6 million particles from four batches were extracted with Relion v.3.1.0 (ref. 88) and processed separately in cryoSPARC v.4.4.0 (refs. 89,90). For each batch, an initial volume was generated and homogeneously refined using three to four 2D classes that best resembled a dimer. It then served as a reference for 3D classification with ten classes. The single best classes of each batch were combined and sorted by four iterations of 2D classification and particle selection. In a final, non-uniform refinement of the selected 118,791 particles, a map of 5.7 Å overall resolution was generated. After auto-sharpening, a resolution of ~4.7 Å was reached. Local resolution estimation in cryoSPARC was performed using 1,530,500 voxels with a local box size of 28. Dimer flexibility was analyzed with '3DFlex'[53] in cryoSPARC. A mesh of 20 tetra cells was trimmed to the contours of the previous consensus volume and the algorithm trained with

two latent dimensions. Deformation was visualized by generating a 41-frame volume series through the latent space and rendering in ChimeraX v.1.6.1 (ref. 91). An AF2 model of the dimer was docked into the map using Phenix v.1.20.1-4487 (ref. 92) and adjusted by simulated annealing. Refinement was performed with phenix.real_space_refine, Coot v.0.9.6 (ref. 93) and the ISOLDE[94] module of ChimeraX[91]. Loop 1 (molecule A, residues 337–349; molecule B, residues 335–351), loop 2 (molecules A and B, residues 384–442), the C-terminus of molecule A (residues 904–909) and the C-lobe of molecule B (residues 788–909) were removed. Structural illustrations were created with PyMol v.2.5.0 (Schrödinger) or ChimeraX.

**HACE1 ΔN–RAC1 Q61L complex.** The SEC-purified SIA-linked complex was crosslinked at a concentration of 4 µM with 0.1 mM BS3 (bis(sulfosuccinimidyl)suberate) at room temperature for 30 min. After quenching with 200 mM Tris (pH 6.5), 4 µl was applied to glow-discharged R1.2/1.3 Cu300 grids (Quantifoil). Grids were treated as above but with a blotting force of 5 for 7 s. Data collection was performed as above but with a ~30° stage tilt and an overall electron dose of 60 e⁻ Å⁻² per image. After motion correction, dose weighting, contrast-transfer function estimation and particle picking in Warp, seven million particles collected from two grids and 34,369 micrographs were extracted with Relion in six batches and processed in cryoSPARC. For a subset of 600,000 particles, 2D classes were generated and classes representing the complex were selected for ab initio volume generation. Likewise, two separate batches of 'junk' classes were selected to create 'junk volumes'. The volumes were used for guided 3D classification via heterogenous refinement in three cycles, in which the particles of the complex were used as input for the next refinement round. The batches were gradually merged with each cycle. The output volume and particles of a non-uniform refinement, with per-particle defocus optimization to account for the stage tilt, served as a reference for 3D classification with five classes. The single best class with 256,595 particles resulted in a 4.2 Å resolution map after a final, non-uniform refinement. Local resolution estimation was performed with cryoSPARC, using 1,023,546 voxels with a local box size of 26. The final map was filtered based on the map 'local resolution estimation'. A HACE1–RAC1 model, extracted from an AF2 prediction of a HACE1–RAC1–Ub complex was docked into the map with Phenix. Refinement was performed as described above. Loop 1 (residues 339–348), loop 2 (residues 396–436), the C-terminal region of HACE1 (residues 904–909) and residues 179–192 of RAC1 were removed. A crystal structure of Gpp(NH)p-bound RAC1 (PDB 1MH1 (ref. 95)) provided a template for the positioning of GTP. A cif-file for the SIA-crosslink was generated with Phenix eLBOW.

### AF2
Structures of HACE1 (monomer) were predicted with AlphaFold v.2.3.1 *monomer_ptm* (https://github.com/deepmind/alphafold)[96]; those of a HACE1 dimer and a HACE1–RAC1–Ub complex were predicted with AlphaFold v.2.3.1 *multimer*[97]. The PAE plots were rendered with ChimeraX[98].

### SAXS
SEC–SAXS data of HACE1 FL and ΔN were collected at beamline P12 of the Deutsches Elektronensynchrotron (DESY, Hamburg, Germany) at an injection concentration of 9.73 mg ml⁻¹ in 50 mM HEPES (pH 8.0), 50 mM NaCl and 5 mM DTT. Data were processed with CHROMIXS, PRIMUS, AUTORG and GNOM, as implemented in ATSAS v.3.0.5 (ref. 99). Based on the distance distribution, an envelope was generated with GASBOR v.2.3 (ref. 56) and superimposed with the cryo-EM structure using SUPCOMB[100]. For the simulation of HACE1 FL scattering with AllosMod-FoXS[54,55], we input the cryo-EM structure of the dimer, determined here, upon modeling of the second C-lobe (molecule B) in an inverted-T conformation. Missing loops were modeled with AllosMod-FoXS, using default settings and sampling of the most probable conformations consistent with the input structure. The best-scoring simulation and fit statistics are reported. For simulations of HACE1 ΔN scattering, we used MultiFoXS[54]. As input structure, molecule A was extracted from the structure of the dimer, determined here, residues 1–21 were removed and missing loops built with AllosMod-FOXS. The best-scoring model with a c2-value of <2 was input into MultiFoXS, using default settings; two hinges were defined flexible (residues 767–771 (inter-lobe linker of the HECT domain) and residues 501–505 (hinge between the ANKs and the HECT N-lobe)).

### HDX–MS
The following concentrations were used: 50 µM HACE1 *apo* (Supplementary Data 1); 50 µM RAC1 Q61L ±100 µM HACE1 ΔN or 50 µM HACE1 ΔN ±100 µM RAC1 Q61L (Supplementary Data 2); and 25 µM of HACE1 S14E or I694D ±50 µM RAC1 Q61L (Supplementary Data 3). A total of 58.5 µl of D₂O-based buffer (Supplementary Data 1: 20 mM HEPES (pH 8.0), 150 mM NaCl, 5 mM DTT; Supplementary Data 2 and 3: 20 mM HEPES (pH 7.8), 100 mM NaCl, 1 mM MgCl₂, 5 mM DTT) was added to 6.5 µl of protein using a two-arm autosampler (LEAP Technologies)[101]. For additional details, see Supplementary Methods.

### In-vitro ubiquitination assays
Multi-turnover ubiquitination reactions containing 50 nM UBA1 (E1), 0.3 µM UBE2L3 (E2), 0.5 µM HACE1, 5 µM RAC1 Q61L (if applicable) or 2 µM OPTN (if applicable) and 30 µM Ub in 50 mM HEPES (pH 8.0), 100 mM NaCl, 10 mM MgCl₂ and 1 mM TCEP were started by addition of 5 mM ATP and incubated at 30 °C. Single-turnover E2~Ub discharge assays were performed as previously described[18]. In short, 0.5 µM UBA1, 10 µM UBE2L3 and 20 µM Ub were incubated with 2 mM ATP and 10 mM MgCl₂ in 50 mM HEPES (pH 8.0) and 150 mM NaCl at 30 °C for 15 min, diluted fourfold and quenched with 50 mM EDTA. Thereafter, 0.5 µM UBE2L3~Ub was incubated with 2.5 µM HACE1 at 30 °C. A portion of Ub was fluorophore-labeled. Reactions were quenched with SDS-loading dye at the indicated times and analyzed by SDS–PAGE, fluorescence scanning (Odyssey CLx; LI-COR) and Coomassie staining. In the fluorescence images, only marker bands are labeled that are visible by fluorescence, whereas all bands are labeled in the Coomassie-stained images.

### Mammalian cell culture
HeLa Kyoto cells (provided by Peter Lenart, MPI NAT) were cultured in DMEM, high-glucose, pyruvate (Thermo Fisher Scientific) with 10% (v/v) FBS (Thermo Fisher Scientific), 1% (v/v) penicillin-streptomycin (Sigma-Aldrich), according to standard techniques at 37 °C in 5% CO₂ and were regularly tested for mycoplasma contaminations. For transient transfections, cells at 70% confluency were treated with Lipofectamine 3000 (Thermo Fisher Scientific) in FBS-free media. FBS-containing media was used 4 h after transfection. The collected cells were resuspended in 50 mM HEPES (pH 8.0), 150 mM NaCl, 1% (v/v) Triton X-100, 1.5 mM MgCl₂, protease–phosphatase inhibitor cocktail (Sigma-Aldrich) 24 h after transfection, incubated on ice for 20 min and the lysate cleared by centrifugation. Total protein concentrations of the cleared lysates were determined with BCA assays (Pierce).

### Generation of HEK293 Flp-In T-REx stable cell lines
Stably transfected HEK293 cell lines for expression of mClover-tagged FL, ΔN, L15N HACE1 or mClover under the control of a tetracycline-inducible promoter were generated with the Flp-In T-REx system (Thermo Fisher Scientific). In brief, cells were co-transfected with the respective plasmid and pOG44 (encoding the Flp-recombinase) at a 1:3 ratio. Then, 100 µg ml⁻¹ hygromycin B was applied for selection, foci of resistant cells were pooled and HACE1 expression was confirmed by immunoblotting and fluorescence microscopy.

## Antibodies

The following primary antibodies were used: anti-HA mouse monoclonal antibody (H9658, Sigma-Aldrich; dilution 1:10,000); anti-GFP rabbit antiserum (132002, Synaptic Systems; 1:1000); anti-HACE1 rabbit monoclonal antibody (EPR7962, ab133637, Abcam; 1:500); and anti-Ub mouse monoclonal antibody P4D1 (sc-8017, Santa Cruz Biotechnology; 1:1,000). Fluorescently labeled donkey secondary antibodies included anti-mouse IRDye 680RD and anti-rabbit IRDye 800CW (926-68072 and 926-32213, LI-COR; 1:20,000). For luminescence-based detection, goat anti-rabbit HRP-linked antibody 7074 (Cell Signaling Technology; 1:10000) was used.

## Cell-based IPs

For co-IPs, 5 µg each of pcDNA3.1+HA and CC037_pCMV-mClover-GW, encoding HACE1 variants, were co-transfected into HeLa Kyoto cells. Cleared lysates were diluted to 4 µg µl$^{-1}$ protein in 30 mM HEPES (pH 8.0), 150 mM NaCl, 0.5 mM EDTA and protease–phosphatase inhibitor cocktail (Sigma-Aldrich) and incubated with 25 µL GFP-Trap magnetic agarose (ChromoTek) at 4 °C for 1 h. The resin was washed three times with the same buffer, including 0.1% (v/v) Igepal CA-630 (Sigma-Aldrich) and proteins were eluted in 40 µl of 2× SDS-loading dye at 95 °C for 5 min. For immunoblotting, 5 µl of input (1%) and 10 µl of eluate (20%) were subjected to SDS–PAGE, transferred to an immobilon-FL PVDF membrane (Sigma-Aldrich), blocked with 5% (w/v) BSA in TBS-T (20 mM Tris (pH 7.6), 150 mM NaCl, 0.1% (v/v) Tween-20) and incubated with primary antibody overnight. Fluorescently labeled secondary antibodies were used for detection (Li-COR). Co-IPs using the leaky expression of stable HEK293 cell lines were conducted similarly, incubating 5 µg µl$^{-1}$ protein with GFP-Trap resin and subjecting 20 µl (1.6%) of input and 5 µl of eluate (14%), respectively, to SDS–PAGE.

To monitor ubiquitination, the transfection, IP and immunoblotting protocols were similar to those outlined above. A total of 5 µg of pcDNA3.1+HA encoding HACE1, pcDNA3-EGFP-RAC1(T17N) or pcDNA3-EGFP-RAC1 Q61L, respectively, were transfected into HeLa Kyoto cells and cells treated with 10 µM MG-132 (Sigma-Aldrich) 4 h before lysis. The lysis buffer was the same as for IPs but contained 10 µM MG-132 and 5 mM N-ethylmaleimide (Sigma-Aldrich). For IPs, lysates were diluted to ~3 µg µl$^{-1}$ protein. Elution was performed with 35 µl of 2× SDS-loading dye at 95 °C for 5 min. For immunoblotting, 20 µg of the input and 15 µl of the eluate (43%) were analyzed by SDS–PAGE.

## Native PAGE analyses

Mouse and rat brain homogenates (provided by Reinhard Jahn, MPI NAT), originated from Sprague Dawley rats and C57BL/6 mice maintained at the MPI NAT, according to international animal welfare rules (Federation for Laboratory Animal Science Associations guidelines and recommendations). Homogenates were prepared as previously described[102]. In short, fresh brains were washed with 320 mM sucrose, 5 mM HEPES pH 7.3 and homogenized in the same buffer with 240 µM PMSF and 1 µg ml$^{-1}$ pepstatin (1 ml buffer per gram of brain tissue), using a douncer. The crude homogenate was spun at 730×g at 4 °C for 10 min and the total protein concentration of the supernatant was determined by BCA assays before snap-freezing and storage at −80 °C.

For native PAGE analyses, samples were supplemented with native Tris-glycine sample buffer (Novex, Thermo Fisher Scientific) and loaded onto a 4–8% Tris-acetate gel (Nu-PAGE, Thermo Fisher Scientific) (~20 µg of the brain homogenates; ~2 ng of recombinant controls). Proteins were transferred to a nitrocellulose membrane (Protran 0.45 µM, GE Healthcare) in 25 mM Tris, 192 mM glycine, 15% (v/v) MeOH and 0.01% (w/v) SDS, the membrane blocked with 5% (w/v) BSA in TBS-T (20 mM Tris (pH 7.6), 150 mM NaCl, 0.1% (v/v) Tween-20) and incubated with anti-HACE1 antibody overnight. For detection, a horseradish peroxidase-labeled secondary antibody, SignalFire ECL reagent (Cell Signaling Technology) and a Fujifilm LAS-1000 imaging system were used.

## Statistics and reproducibility

All gel-based analyses, including in-vitro and cell-based activity assays and IPs, of which representative results are displayed, were independently repeated at least three times with similar results.

## Reporting summary

Further information on research design is available in the Nature Portfolio Reporting Summary linked to this article.

## Data availability

The cryo-EM structures of HACE1 FL and the HACE1 ΔN–RAC Q61L complex were deposited under PDB 8PWL and 8Q0N and the maps under IDs EMD-17994 and EMD-18056, respectively. The SAXS data for HACE1 FL and HACE1 ΔN have been deposited under SASBDB IDs SASDTC5 and SASDTD5, respectively. The HDX–MS data were deposited to the ProteomeXchange Consortium via the PRIDE partner repository, ID PXD045837. Source data are provided with this paper.

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

## Acknowledgements

We acknowledge funding from the SFB1190 (ID 264061860, P24N; German Research Foundation (DFG); to S.L.), iNEXT Discovery (ID 25363; to S.L.), the MPI NAT (to S.L.), the Max Planck Society (to S.L.), the EMBO Young Investigator Program (to S.L.), the EMBO Postdoctoral Program (ID EMBO ALTF 439-2022; to M.W.) and SFB1565 (ID 469281184, P12; DFG; to K.E.B.). We thank P. Cramer (MPI NAT) for access to the in-house cryo-EM facility and the DFG for co-financing the HDX–MS instrumentation (ID 260989694; Marburg); R. Jahn and M. Ganzella (MPI NAT) for brain homogenates; C. Blanchet (DESY), P. Pohl (MPI NAT) and S. Dennerlein (University Medical Center Göttingen) for expert advice; K. Remans and A. Börgel (European Molecular Biology Laboratory, Protein Expression and Purification Core Facility, Heidelberg, Germany) for HACE1 expression in Sf9 and HEK293F cells; N. Ranjan and U. Steuerwald (MPI NAT) for technical assistance; and all colleagues who kindly provided plasmids to us.

The funders had no role in study design, data collection and analysis, decision to publish or preparation of the manuscript.

## Author contributions

J.D., M.W. and J.J.T. conducted protein preparations and in-vitro assays. J.D., M.W. and C.D. performed cryo-EM analyses. J.D., J.J.T. and C.T. performed cell-based assays. C.T. and K.E.B. prepared cell lines. M.W. conducted crosslinking analyses. J.D. and S.L. performed SAXS. W.S. performed HDX–MS. O.D. and H.U. performed MS. T.J.F. performed AF2 analyses. S.L. designed the study and prepared the paper, with input from all others.

## Funding

## Competing interests

The authors declare no competing interests.

## Additional information

**Extended data** is available for this paper at https://doi.org/10.1038/s41594-023-01203-4.

**Correspondence and requests for materials** should be addressed to Sonja Lorenz.

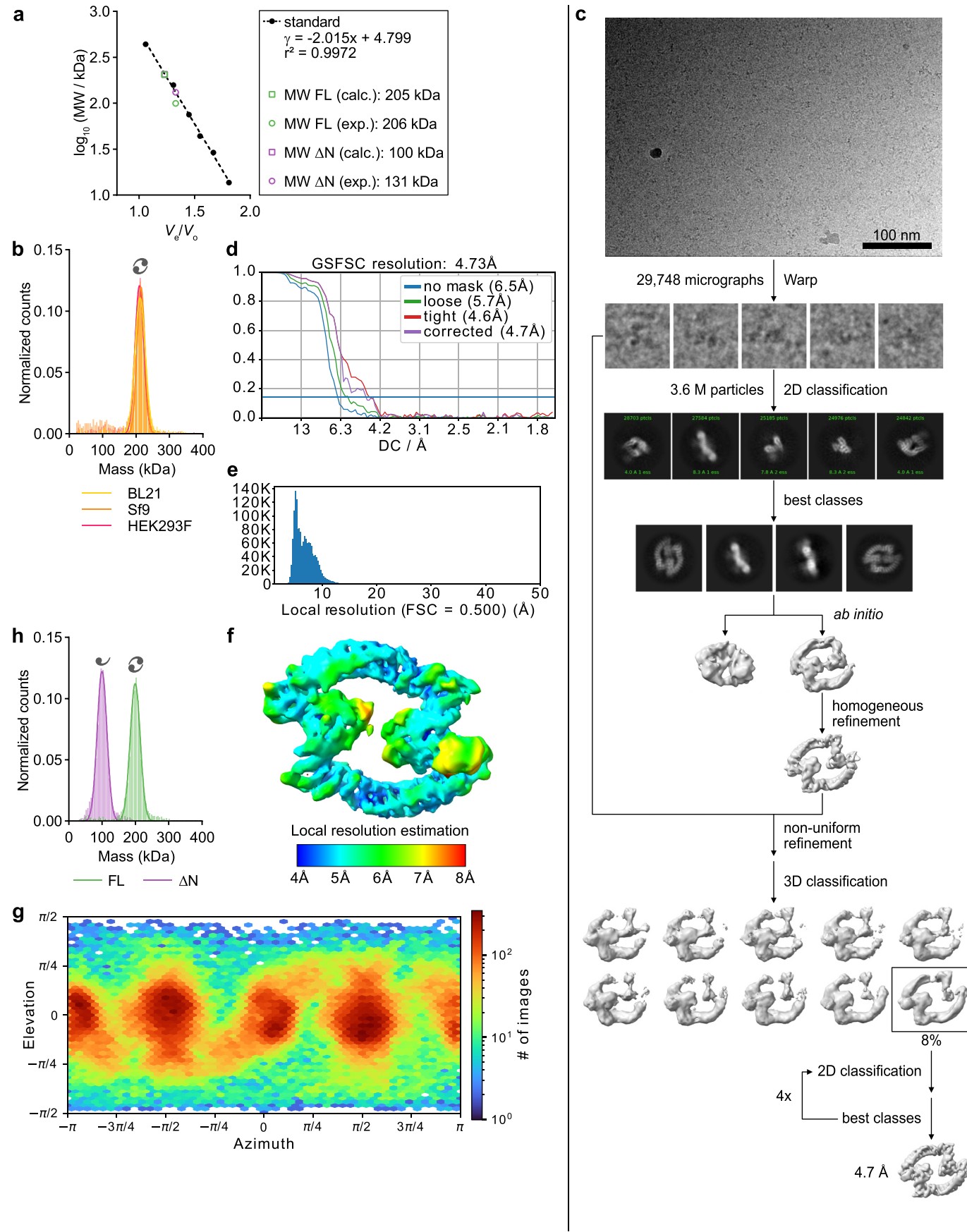

**Extended Data Fig. 1 | See next page for caption.**

**Extended Data Fig. 1 | SEC, MP, and cryo-EM analyses of HACE1. (a)** SEC-based MW estimation of HACE1 FL and ΔN, based on a calibration curve of globular proteins (ferritin (440 kDa), aldolase (158 kDa), conalbumin (75 kDa), ovalbumin (44 kDa), carbonic anhydrase (29 kDa), and ribonuclease A (13.7 kDa) (Cytiva)); $V_e$: elution volume; $V_0$: void volume (left); the linear fit equation and the experimentally derived ('exp.') and calculated ('calc.') MW values are provided (right). **(b)** Comparative MP analyses of HACE1 FL, expressed in *E. coli* BL21, Sf9, and HEK293F cells, respectively; the MWs and other parameters are provided in Extended Data Table 1. **(c)** Cryo-EM processing scheme for HACE1 FL. A representative raw micrograph, 2D classes, and intermediate processing steps are shown. **(d)** Fourier shell correlation (FSC) plots for the HACE1 FL dimer reconstruction and resolution estimation, as reported by cryoSPARC[89]. **(e)** Histogram of local resolution estimation at a FSC threshold of 0.5. **(f)** Local resolution-filtered map of the HACE1 FL reconstitution, colored as indicated. **(g)** Angular distribution plot, with blue representing a low and red a high number of particles per angular bin. **(h)** MP analysis of HACE1 FL and ΔN at a reduced ionic strength (50 mM NaCl) compared to Fig. 1a (150 mM NaCl). For MP-derived MWs and other parameters, see Extended Data Table 1.

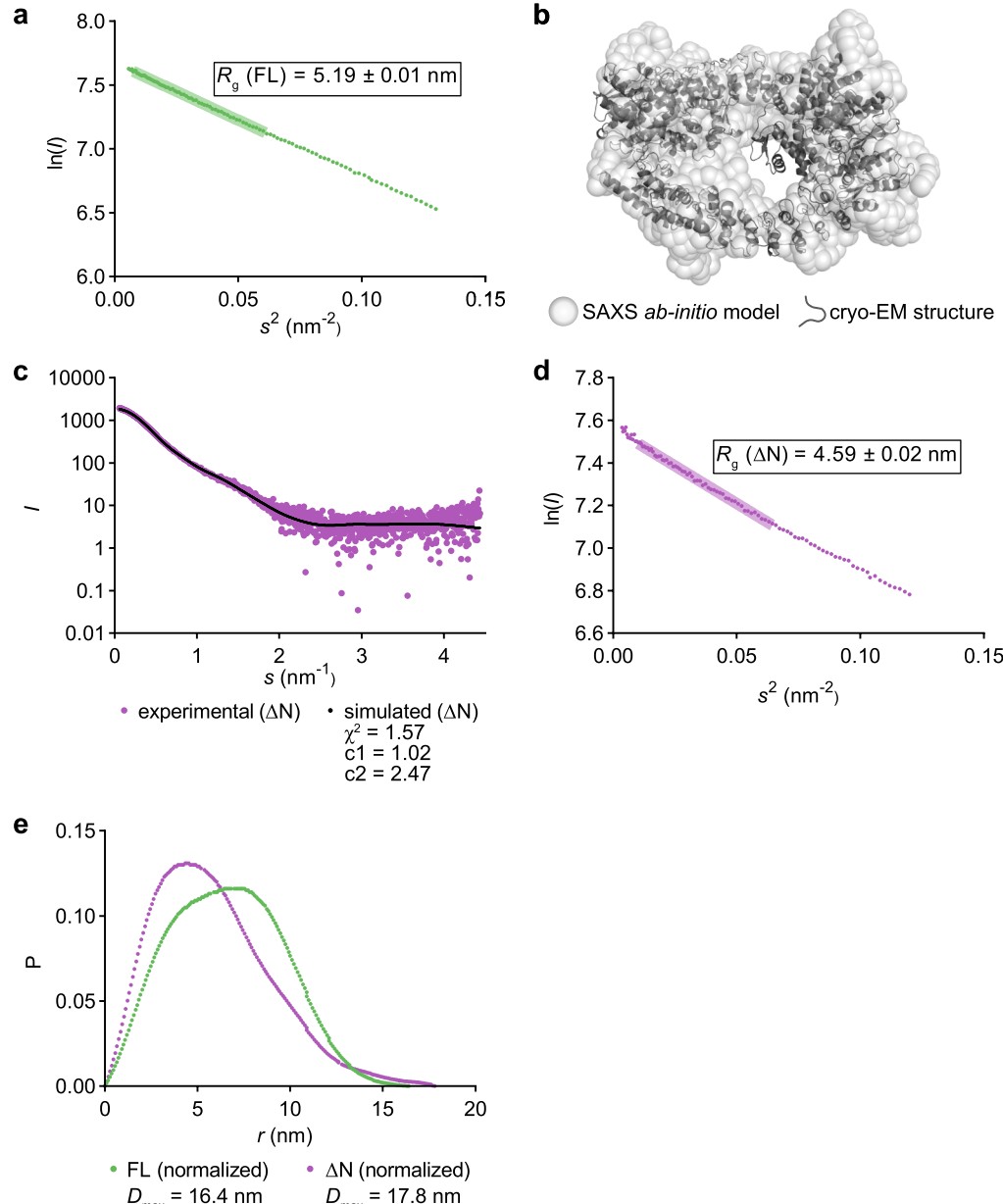

**Extended Data Fig. 2 | SAXS analyses of HACE1 FL and ΔN. (a)** Guinier plot for HACE1 FL from Fig. 1e with linear fit (pale, thick line). The error is defined as the standard deviation of experimental data from the fit in the shown interval, plus the standard deviation of $R_g$-values from all possible intervals from the $R_g$ of the selected one, calculated with AUTORG[99]. **(b)** SAXS-based ab-initio structure reconstruction of HACE1 FL, generated with GASBOR[56]; the cryo-EM structure of HACE1 FL (with the second C-lobe modelled in an inverted T-shape; missing loops not modelled) was fitted into the SAXS-derived envelope with SUPCOMB[100]. **(c)** SAXS data of HACE1 ΔN ($I$ = scattering intensity; $s$ = momentum transfer) (purple), superposed with a simulated curve (black); the simulation was generated with MultiFoXS[54], based on molecule A (residues 22-902) of the cryo-EM structure and modeling of missing regions with AllosMod-FoXS[54,55]. The MultiFoXS fitting parameters are provided, $R_g$-values in Fig. 1f. The misfit at low angles suggests the particle is more elongated in solution than the input structure. **(d)** Guinier plot for HACE1 ΔN from (c) with linear fit (pale, thick line), calculated with AUTORG. Errors are defined as in (a). **(e)** SAXS-derived pair distance distribution functions for HACE1 FL and ΔN, calculated with GNOM; the maximal dimension, $D_{max}$, are provided.

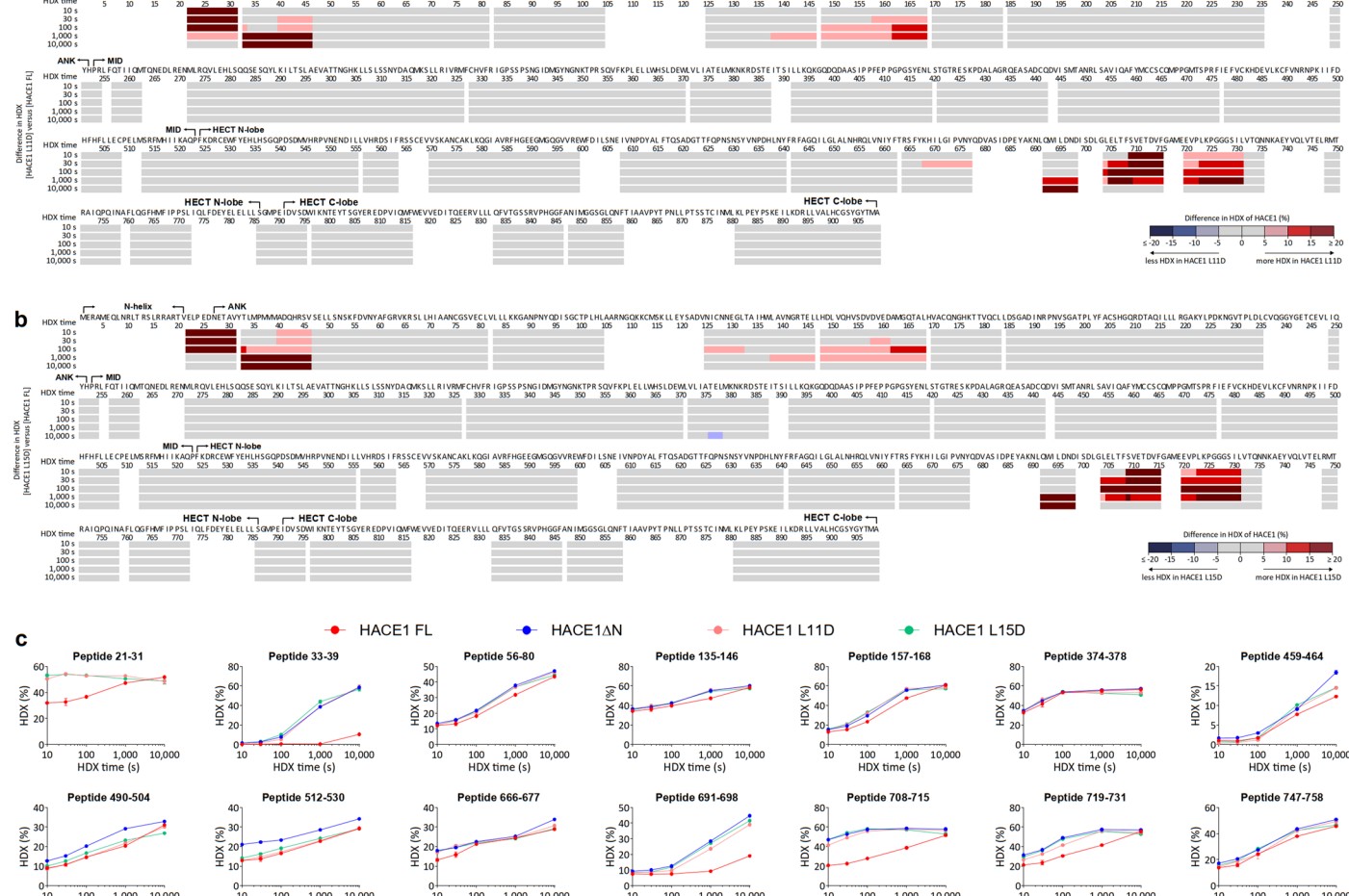

**Extended Data Fig. 3 | HDX-MS analyses of HACE1 wild type, L11D, and L15D.**
(a) HDX differences between HACE1 L11D and wild type ('FL') at the indicated time points, shown along the amino acid sequence; domain boundaries are indicated. For raw data, see Supplementary Data 1. (b) HDX differences between HACE1 L15D and wild type ('FL') at the indicated time points, shown as in (a). (c) Extent of HDX monitored over time for representative peptides derived from HACE1 variants, plotted as the mean and standard deviations of 3 technical replicates. ΔN is included for comparison.

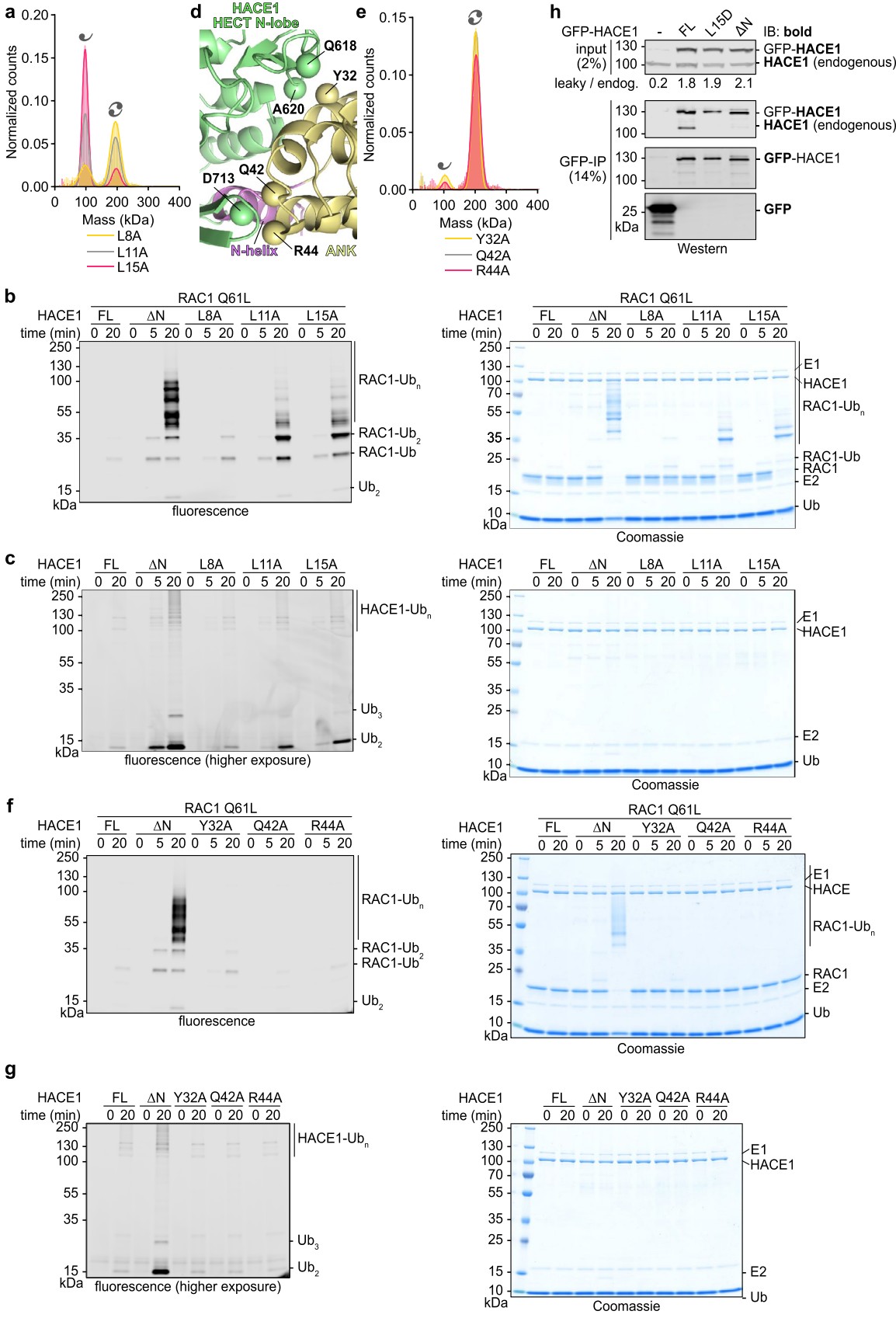

**Extended Data Fig. 4 | See next page for caption.**

**Extended Data Fig. 4 | Characterization of dimer-interface variants of HACE1. (a)** MP analysis of full-length HACE1 variants; for MWs and other parameters, see Extended Data Table 1. **(b)** Reconstituted multi-turnover ubiquitination assay, monitoring the activity of HACE1 variants toward RAC1 Q61L; ubiquitinated products are visualized by fluorescence imaging (left), before Coomassie staining (right). FL = wild type. **(c)** Reconstituted multi-turnover assay, monitoring autoubiquitination and free Ub chain formation by HACE1 variants, analogously to (b); autoubiquitination is less efficient than RAC1 ubiquitination, necessitating a lower intensity threshold compared to (b) ('higher exposure'). Ubiquitinated products are visualized by fluorescence imaging (left), before Coomassie staining (right). FL = wild type. **(d)** Expanded view of a peripheral region of the subunit interface in the cryo-EM structure of HACE1 FL, highlighting intermolecular contacts of Tyr32, Gln42 and Arg44 of ANK1 with residues of the HECT N-lobe; due to the limited resolution of the structure, only $C^{\beta}$-atoms (spheres) are displayed, instead of full side chains. **(e)** MP analysis of full-length HACE1 variants; for MWs and other parameters, see Extended Data Table 1. **(f)** Reconstituted multi-turnover ubiquitination assay, monitoring the activity of HACE1 variants toward RAC1 Q61L; ubiquitinated products are visualized by fluorescence imaging (left), before Coomassie staining (right). FL = wild type. **(g)** Reconstituted multi-turnover assay, monitoring autoubiquitination and free Ub chain formation by HACE1 variants, analogously to (f). **(h)** Co-IP, monitoring the association of GFP(mClover)-tagged HACE1 variants with endogenous HACE1 in stably transfected HEK293 Flp-In T-REx cell lines without inducer. The monitored antigen is marked in bold. The densitometric ratio of tagged HACE1 ('leaky') to endogenous ('endog.') ligase is provided. A weak contaminant band is seen at the height of GFP-HACE1 in the GFP-only expressing control lane, giving rise to a ratio of 0.2.

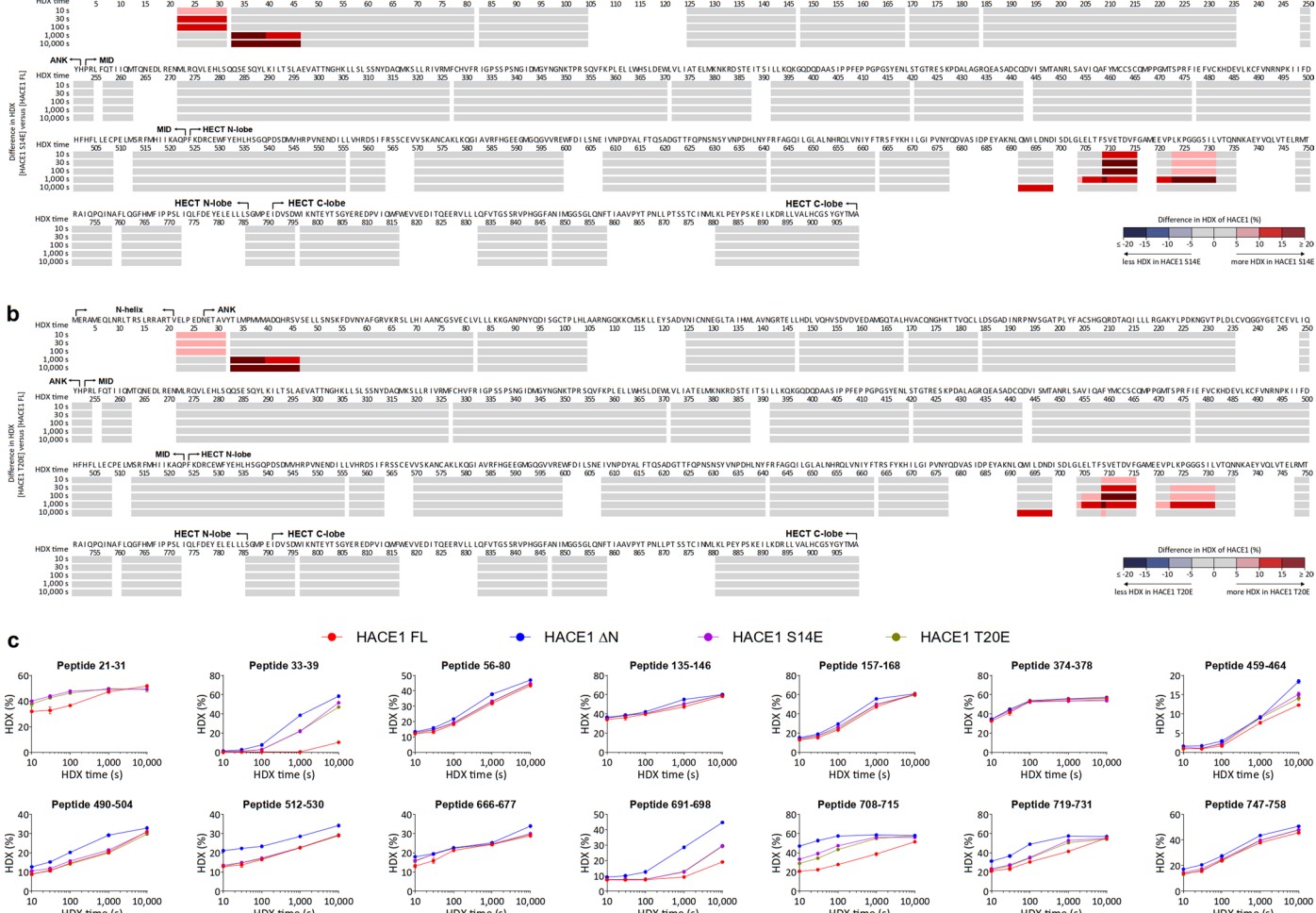

**Extended Data Fig. 5 | HDX-MS analyses of HACE1 wild type, S14E, and T20E.**
(a) HDX differences between HACE1 S14E and wild type ('FL') at the indicated time points, shown along the amino acid sequence; domain boundaries are indicated. For raw data, see Supplementary Data 1. (b) HDX differences between HACE1 T20E and wild type ('FL') at the indicated time points, shown as in (a). (c) Extent of HDX monitored over time for representative peptides derived from HACE1 variants, plotted as the mean and standard deviations of 3 technical replicates. HACE1 ΔN is included for comparison.

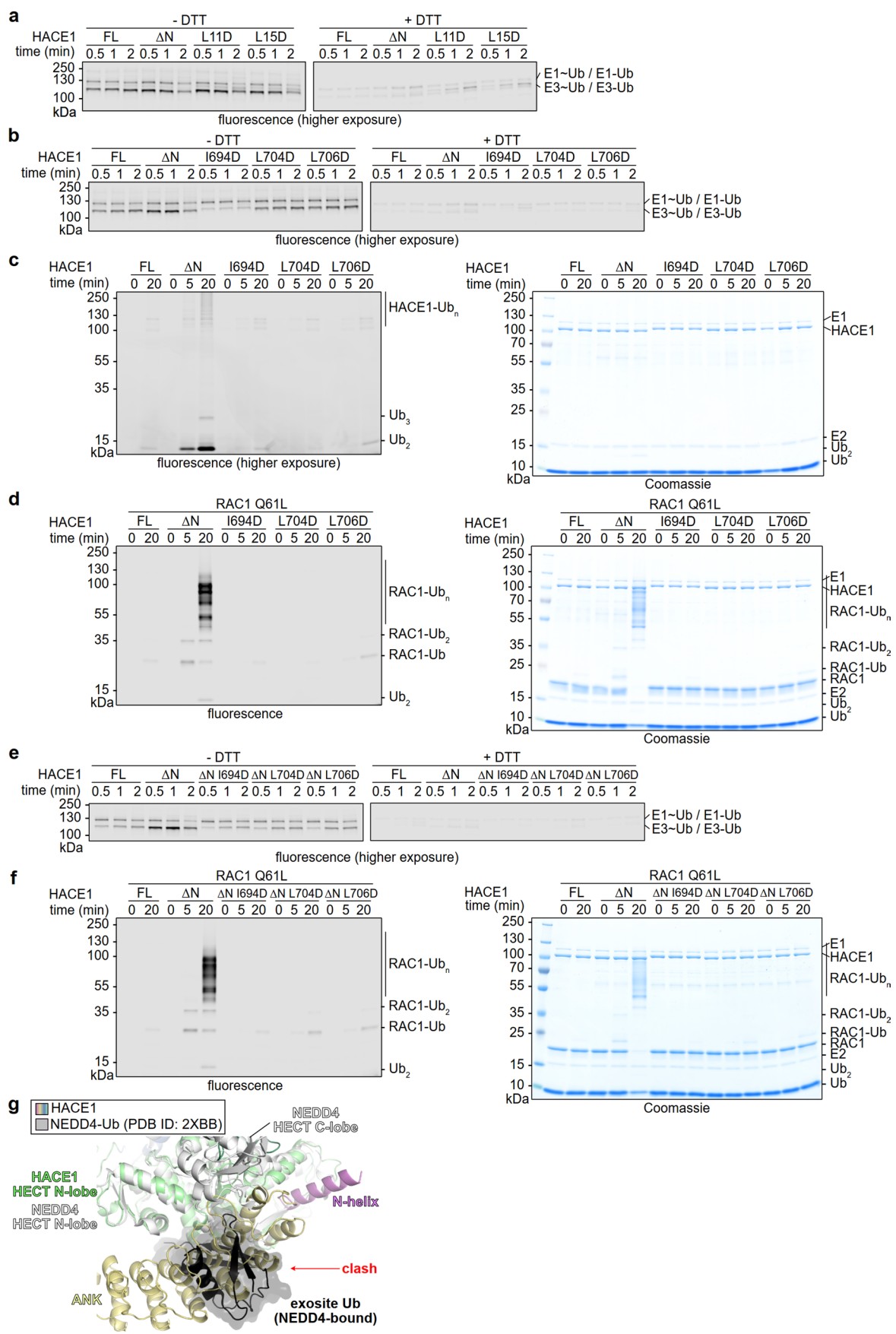

**Extended Data Fig. 6 | See next page for caption.**

**Extended Data Fig. 6 | Mechanistic analyses of HACE1 auto-inhibition.**
**(a)** E3-containing region of the gel from Fig. 4b upon fluorescence imaging; a lower intensity threshold was applied here to visualize thioester ('~') and isopeptide-linked ('·') species ('higher exposure'). FL = wild type. **(b)** E3-containing region of the gel from Fig. 4d upon fluorescence imaging; a lower intensity threshold was applied here to visualize thioester ('~') and isopeptide-linked ('·') species ('higher exposure'). FL = wild type. **(c)** Reconstituted multi-turnover assay, monitoring HACE1 autoubiquitination and free Ub chain formation by SDS PAGE; ubiquitinated products are visualized by fluorescence imaging (left), followed by Coomassie staining (right). Autoubiquitination is less efficient than RAC1 ubiquitination, necessitating a lower intensity threshold compared to (d) ('higher exposure'). FL = wild type. **(d)** Reconstituted multi-

turnover ubiquitination assay, monitoring HACE1 activity toward RAC1 Q61L by SDS PAGE, analogously to (c). FL = wild type. **(e)** E3-containing region of the gel shown in Fig. 4e upon fluorescence imaging; a lower intensity threshold was applied here ('higher exposure'). FL = wild type. **(f)** Reconstituted multi-turnover ubiquitination assay, monitoring HACE1 activity toward RAC1 Q61L by SDS PAGE; ubiquitinated products are visualized by fluorescence imaging (left), followed by Coomassie staining (right). FL = wild type. **(g)** Expanded view of the cryo-EM structure of the HACE1 dimer, superposed with a crystal structure of the NEDD4 HECT domain in complex with exosite-bound Ub (PDB ID: 2XBB ref. [61]); the HECT N-lobes were superposed. Ub is shown as a cartoon and surface, highlighting the clashing of exosite-bound Ub with the HACE1 dimer.

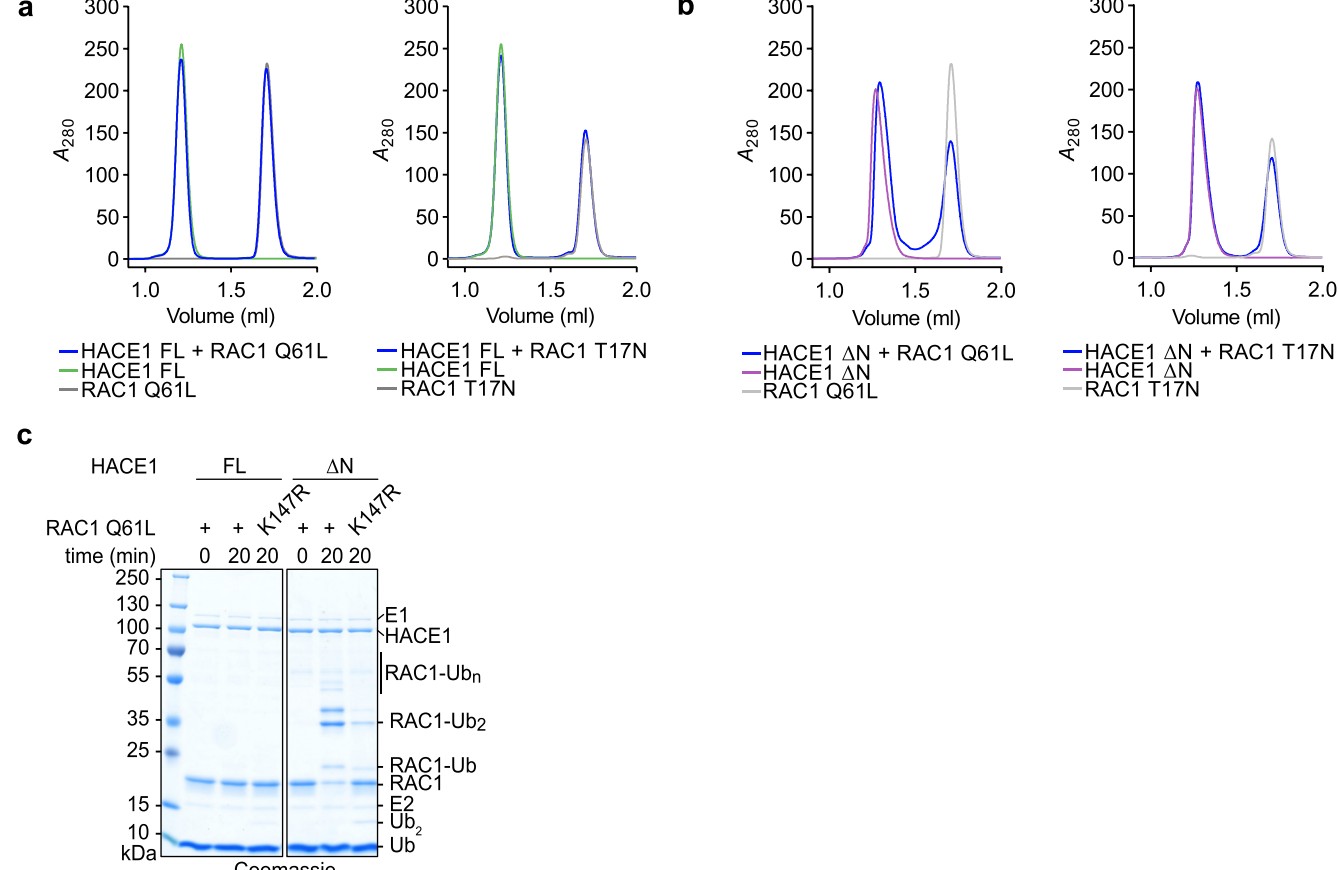

**Extended Data Fig. 7 | SEC analyses of the HACE1-RAC1 interaction. (a)** SEC analyses of HACE1 FL, active RAC1 Q61L (left) and inactive RAC1 T17N (right), respectively, alone or mixed; for details, see Methods. **(b)** SEC analyses of HACE1 ΔN, RAC1 Q61L (left) and RAC1 T17N (right), respectively, alone or mixed; for details, see Methods. **(c)** Reconstituted multi-turnover ubiquitination assay, monitoring the activity of HACE1 variants toward RAC1 variants, monitored by Coomassie staining (bottom).

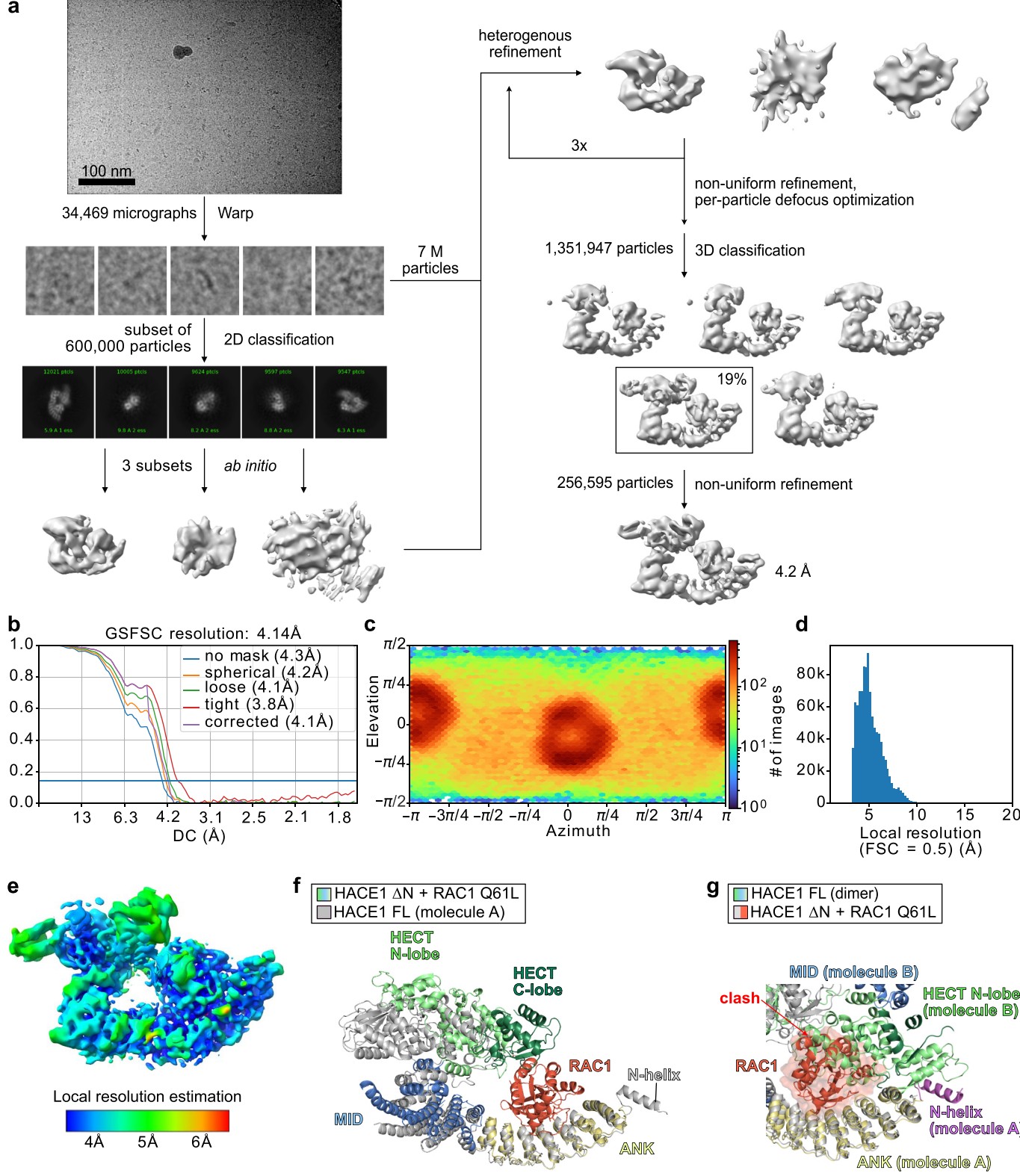

**Extended Data Fig. 8 | Cryo-EM analyses the HACE1 ΔN-RAC1 Q61L complex.**
**(a)** Cryo-EM processing scheme of the HACE1 ΔN-RAC1 Q61L complex; a
representative raw micrograph, 2D classes, and intermediate processing steps
are shown. **(b)** FSC plots for the reconstruction of the complex and resolution
estimation, as reported by cryoSPARC. **(c)** Angular distribution plot, with blue
representing a low and red a high number of particles per angular bin.
**(d)** Histogram of local resolution estimation at a FSC threshold of 0.5. **(e)** Local
resolution-filtered map of the complex reconstitution, colored as indicated.

**(f)** Expanded view of the cryo-EM structures of the HACE1 FL dimer (molecule
A; grey) and the HACE1 ΔN-RAC1 Q61L complex (colored), determined here,
superposed on the ANKs, highlighting the different arrangements of the MID
and HECT domain. **(g)** Expanded view of the cryo-EM structures of the HACE1 FL
dimer (colored as in Fig. 1d) and the HACE1 ΔN-RAC1 Q61L complex (HACE1 in
white and RAC1 in orange), determined here, superposed on the ANKs; RAC1 is
shown in cartoon and surface representations, highlighting the clashing of RAC1
binding with HACE1 dimerization.

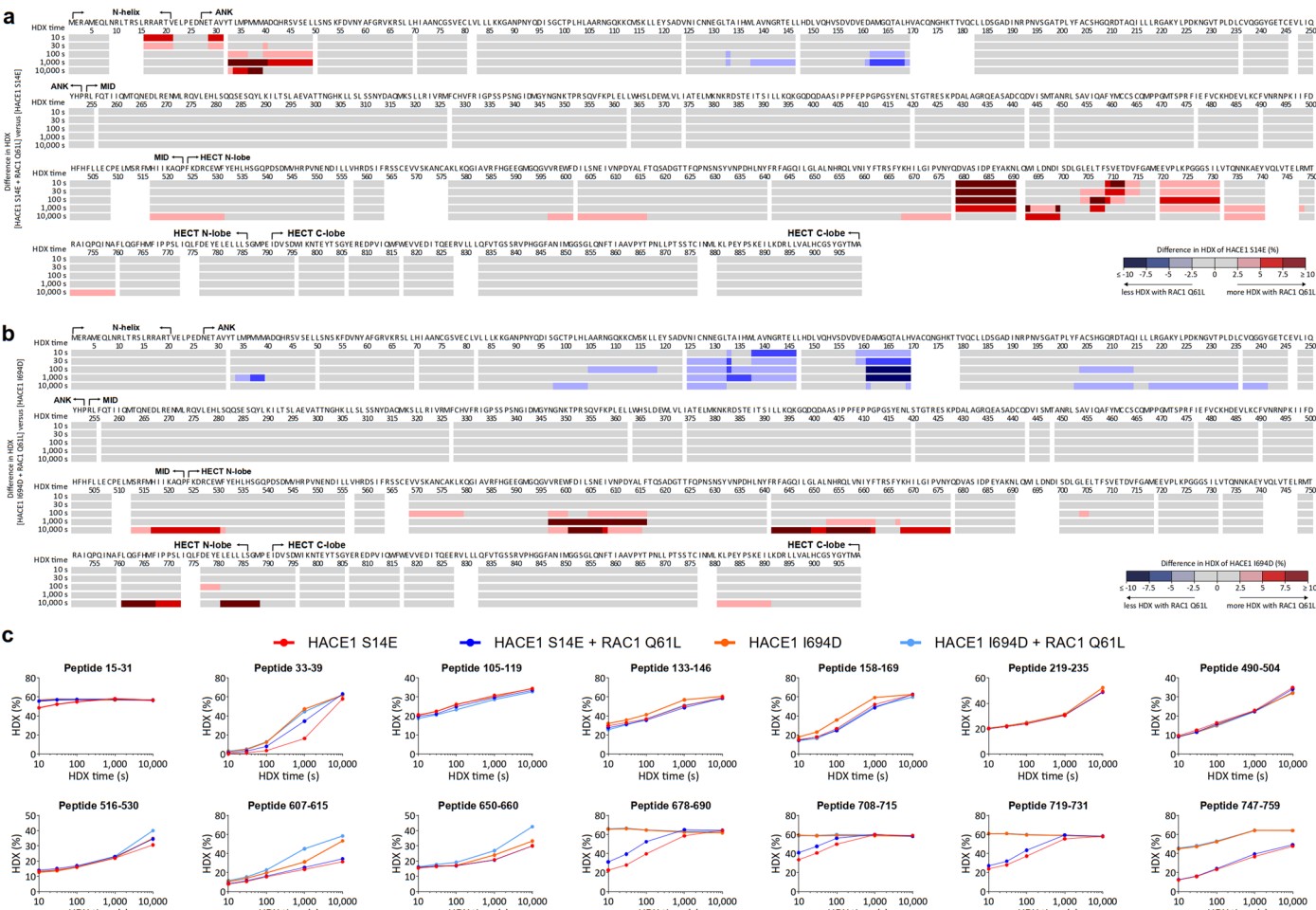

**Extended Data Fig. 9 | HDX-MS analyses of HACE1 variants +/− RAC1 Q61L. (a)** HDX of HACE1 S14E in the absence and presence of RAC1 Q61L, respectively, at the indicated time points, shown along the amino acid sequence; domain boundaries are indicated. For raw data, see Supplementary Data 3. **(b)** HDX of HACE1 I694D in the absence and presence of RAC1 Q61L, respectively, at the indicated time points, shown along the amino acid sequence; domain boundaries are indicated. For raw data, see Supplementary Data 3. **(c)** Extent of HDX monitored over time for representative peptides derived from HACE1 I694D and S14E, respectively, in the absence and presence of RAC1 Q61L.

**Extended Data Table 1 | MP-derived parameters for all analyzed HACE1 samples**

| sample | peak 1 | | | | peak 2 | | | |
|---|---|---|---|---|---|---|---|---|
| | MW (kDa) | σ (kDa) | counts | counts (%) | MW (kDa) | σ (kDa) | counts | counts (%) |
| FL | | | | | 209 | 14 | 8620 | 82 |
| ΔN | 110 | 16 | 8016 | 64 | | | | |
| L8D | 101 | 11 | 2916 | 31 | 199 | 14 | 4594 | 49 |
| L8A | 98 | 18 | 1702 | 22 | 195 | 16 | 4736 | 62 |
| L11D | 105 | 12 | 10227 | 77 | | | | |
| L11A | 98 | 11 | 5638 | 47 | 195 | 14 | 5028 | 42 |
| T12E | 98 | 14 | 1154 | 11 | 204 | 14 | 7857 | 72 |
| S14E | 100 | 9 | 5453 | 74 | 204 | 13 | 496 | 7 |
| L15D | 98 | 12 | 9476 | 84 | | | | |
| L15A | 98 | 9 | 4005 | 74 | 197 | 13 | 757 | 14 |
| T20E | 101 | 8 | 3025 | 70 | 208 | 11 | 998 | 23 |
| Y32A | 104 | 13 | 347 | 8 | 202 | 11 | 3118 | 76 |
| Q42A | | | | | 203 | 12 | 4343 | 85 |
| R44A | 104 | 10 | 244 | 3 | 204 | 14 | 6019 | 79 |
| I694D | 98 | 11 | 7727 | 89 | | | | |
| L704D | 97 | 12 | 647 | 10 | 199 | 12 | 4860 | 75 |
| L706D | 98 | 9 | 6168 | 80 | 199 | 11 | 967 | 12 |
| FL (BL21) | | | | | 213 | 15 | 4209 | 85 |
| FL (Sf9) | | | | | 215 | 12 | 1753 | 69 |
| FL (HECT293F) | | | | | 211 | 13 | 3009 | 80 |
| ΔN + RAC1 | 125 | 13 | 5948 | 80 | | | | |

Missing values indicate that the respective peak was not present. The calculated MWs are 102.4 kDa for a HACE1 FL (wild type) monomer, 99.9 kDa for HACE1 ΔN, and 123.8 kDa for the HACE1 ΔN–RAC1 Q61L complex.

# Reporting Summary

## Statistics

For all statistical analyses, confirm that the following items are present in the figure legend, table legend, main text, or Methods section.

| n/a | Confirmed | |
|---|---|---|
| ☐ | ☒ | The exact sample size (*n*) for each experimental group/condition, given as a discrete number and unit of measurement |
| ☐ | ☒ | A statement on whether measurements were taken from distinct samples or whether the same sample was measured repeatedly |
| ☒ | ☐ | The statistical test(s) used AND whether they are one- or two-sided<br>*Only common tests should be described solely by name; describe more complex techniques in the Methods section.* |
| ☒ | ☐ | A description of all covariates tested |
| ☒ | ☐ | A description of any assumptions or corrections, such as tests of normality and adjustment for multiple comparisons |
| ☐ | ☒ | A full description of the statistical parameters including central tendency (e.g. means) or other basic estimates (e.g. regression coefficient) AND variation (e.g. standard deviation) or associated estimates of uncertainty (e.g. confidence intervals) |
| ☒ | ☐ | For null hypothesis testing, the test statistic (e.g. *F*, *t*, *r*) with confidence intervals, effect sizes, degrees of freedom and *P* value noted<br>*Give P values as exact values whenever suitable.* |
| ☒ | ☐ | For Bayesian analysis, information on the choice of priors and Markov chain Monte Carlo settings |
| ☒ | ☐ | For hierarchical and complex designs, identification of the appropriate level for tests and full reporting of outcomes |
| ☒ | ☐ | Estimates of effect sizes (e.g. Cohen's *d*, Pearson's *r*), indicating how they were calculated |

*Our web collection on statistics for biologists contains articles on many of the points above.*

## Software and code

Policy information about availability of computer code

| Data collection | AcquireMP version 2023_R2.2 with IScat v1.58.0 and IScat Utils v 1.45.0 (Refeyn), SerialEM version 4.0 (University of Colorado Boulder), Image Reader LAS-1000 Pro version 2.6 (Fujifilm) |
|---|---|
| Data analysis | DiscoverMP version 2023_R2 with IScat v1.58.0 and IScat Utils v 1.45.0 (Refeyn), PhotoMol accessed 2023-09-05 (EMBL), Prism version 9.5.1 (733) (GraphPad), Warp version 1.0.9 (Dimitry Tegunov, Patrick Cramer), Relion version 3.1.0 (Sjors Scheres, MRC LMB), CryoSparc version 4.4.0+231114 (Structura Biotechnology), Pymol version 2.5.0 (Schrödinger), Phenix version 1.20.1-4487 (Lawrence Berkeley National Laboratory), ChimeraX version 1.6.1 (UCSF), AlphaFold monomer/multimer version 2.3.1 (DeepMind), ATSAS version 3.0.5 (EMBL), GASBOR version 2.3 (EMBL), SUPCOMB version 13+20 (EMBL), AllosMod-FoXS version main.1d1c5f6 (UCSF), Multi-FoXS version main.26c04d3 (UCSF), ImageStudio Lite version 5.2 (LI-COR) |

For manuscripts utilizing custom algorithms or software that are central to the research but not yet described in published literature, software must be made available to editors and reviewers. We strongly encourage code deposition in a community repository (e.g. GitHub). See the Nature Portfolio guidelines for submitting code & software for further information.

## Data

Policy information about availability of data

All manuscripts must include a data availability statement. This statement should provide the following information, where applicable:

- Accession codes, unique identifiers, or web links for publicly available datasets
- A description of any restrictions on data availability
- For clinical datasets or third party data, please ensure that the statement adheres to our policy

> The cryo-EM structures of HACE1 FL and the HACE1 deltaN-RAC Q61L complex were deposited under PDB IDs 8PWL and 8Q0N; the maps under IDs EMD-17994 and EMD-18056, respectively. The HDX-MS data were deposited to the ProteomeXchange Consortium via the PRIDE partner repository, ID: PXD045837

## Research involving human participants, their data, or biological material

Policy information about studies with human participants or human data. See also policy information about sex, gender (identity/presentation), and sexual orientation and race, ethnicity and racism.

| | |
|---|---|
| Reporting on sex and gender | Not applicable. |
| Reporting on race, ethnicity, or other socially relevant groupings | Not applicable. |
| Population characteristics | Not applicable. |
| Recruitment | Not applicable. |
| Ethics oversight | Not applicable. |

Note that full information on the approval of the study protocol must also be provided in the manuscript.

# Field-specific reporting

Please select the one below that is the best fit for your research. If you are not sure, read the appropriate sections before making your selection.

☒ Life sciences  ☐ Behavioural & social sciences  ☐ Ecological, evolutionary & environmental sciences

For a reference copy of the document with all sections, see nature.com/documents/nr-reporting-summary-flat.pdf

# Life sciences study design

All studies must disclose on these points even when the disclosure is negative.

| | |
|---|---|
| Sample size | The data presented in the manuscript represent the averages and/or representatives of at least 3 independent replicates (see section "statistics and reproducibility: All gel-based analyses, including in-vitro and cell-based activity assays/IPs, of which representative results are displayed, were independently repeated at least 3 times with similar results.") . These sample sizes were chosen to generate data at sufficient depth and assess differences between conditions robustly. These sample sizes are sufficient, since the observed effects of interest are clearly detectable between conditions and robust across replicates. The error bars present in SAXS and HDX MS analyses are detailed in the text. |
| Data exclusions | No data were excluded. |
| Replication | All experiments were performed for at least n=3 independent samples, as described, and all attempts were successful. Immunoblots and enzyme assays were performed independently 3 times with similar results. Biochemical in-vitro experimets and functional cell-based assays were performed on separate and fully independent occassions and verified each other. |
| Randomization | Gel-based samples were run in different orders with the same result. Other than that, randomization is not relevant to this study, as no experimental groups were used. |
| Blinding | MS and structural data were analyzed with script-based pipelines, in which results are largely independent from interference of the researchers. The precise workflows are detailed in the methods section. Gel-based assays were replicated by different individuals. The investigators were not blinded, which is standard in this type of study due to multiple steps that require precise operations for accuracy and precision. |

# Reporting for specific materials, systems and methods

## Materials & experimental systems

| n/a | Involved in the study |
|---|---|
| ☐ | ☒ Antibodies |
| ☐ | ☒ Eukaryotic cell lines |
| ☒ | ☐ Palaeontology and archaeology |
| ☒ | ☐ Animals and other organisms |
| ☒ | ☐ Clinical data |
| ☒ | ☐ Dual use research of concern |
| ☒ | ☐ Plants |

## Methods

| n/a | Involved in the study |
|---|---|
| ☒ | ☐ ChIP-seq |
| ☒ | ☐ Flow cytometry |
| ☒ | ☐ MRI-based neuroimaging |

## Antibodies

| Antibodies used | The following primary antibodies were used: anti-HA mouse monoclonal antibody (H9658; Sigma-Aldrich; dilution 1: 10,000); anti-GFP rabbit antiserum (132002; Synaptic Systems; 1: 1000); anti-HACE1 rabbit monoclonal antibody (EPR7962, ab133637; Abcam; 1: 500); and anti-Ub mouse monoclonal antibody P4D1 (sc-8017; Santa Cruz Biotechnology; 1: 1000). Fluorescently labeled donkey secondary antibodies included anti-mouse IRDye 680RD and anti-rabbit IRDye 800CW (926-68072 and 926-32213; LI-COR; 1: 20,000). For luminescence-based detection goat anti-rabbit HRP-linked antibody 7074 (Cell Signaling Technology; 1: 10000) was used. |
|---|---|
| Validation | Validation was performed by the manufacturer, as follows: ab133637 (abcam): WB; sc-8017 (Santa Cruz): IP, WB, IHC(P), ELISA, IF, FCM; 132 002 (Synaptic Systems): IP, WB, ICC, IHC, IHC(P), EM; H9658 (Sigma-Aldrich): IP, WB, ICC, ELISA; 926-32213 (LI-COR): WB; 926-68072 (LI-COR): WB; #7074 (Cell Signaling Technology): WB |

## Eukaryotic cell lines

Policy information about cell lines and Sex and Gender in Research

| Cell line source(s) | Commercial HeLa Kyoto cells were a gift from Dr. Peter Lenart (MPI NAT, Goettingen, Germany); Sf9, HEK Flp-In, and HEK293F cells were purchased from Thermo Fischer Scientific. |
|---|---|
| Authentication | Cell lines were not authenticated beyond ensuring the presence of known antibiotic resistance markers within their genomes (by growth in the relevant antibiotics), monitoring the morphology and growth rates continuously, and PCR-amplifying the Flp-In locus from genomic DNA extracted from the cells (in case of HEK Flp-In). |
| Mycoplasma contamination | All cell lines were tested negative for mycoplasma contamination. |
| Commonly misidentified lines (See ICLAC register) | No commonly misidentified lines were used. |

