## [Peer Review File · Nature Structural & Molecular Biology]

Peer Review Information

Manuscript Title: Structural mechanisms of autoinhibition and substrate recognition by the ubiquitin ligase HACE1

Corresponding author name(s): Sonja Lorenz

Reviewer Comments & Decisions:

Decision Letter, initial version:
--

Message: 17th Aug 2023

Dear Dr. Lorenz,

Thank you again for submitting your manuscript "Structural mechanisms of autoinhibition and substrate recognition by the ubiquitin ligase HACE1". We now have comments (below) from the 3 reviewers who evaluated your paper. In light of those reports, we remain interested in your study and would like to see your response to the comments of the referees, in the form of a revised manuscript.

You will see that though all referees appreciate the novelty of the structural findings, the support of the structural data by biophysical and functional assays, and the potential mechanistic implications, there are several important issues that need to be addressed in a revised manuscript. More specifically, reviewer #3 requests that you perform additional experiments to adequately support (or disprove) the mechanistic hypothesis that phosphorylation in the N-terminal helix regulates HACE1 dimerisation and activation. Very importantly, reviewer #2 questions the physiological relevance of the findings in the absence of cellular experiments without overexpression which may introduce artefactual monomer/dimer equilibria. We editorially agree that validating that endogenous HACE1 dimerises and providing some quantitative idea at which degree it does so (reviewer #2 point 1) and performing direct HACE1 ubiquitylation assays (reviewer #2 point 2) will be important for the success of the story.

We agree that validating the physiological relevance of your findings with relevant CRISPR knock-in mutants would be best, as reviewer #2 proposes. However, if obtaining such cell-lines would be technically very challenging and out of the timeframe of this work, other cellular systems where the potentially mitigating effects of overexpression can be controlled (e.g. siRNA or shRNA-resistant constructs or other systems) may be informative. At any case, we think that providing further evidence for the dimerisation of

HACE1 under physiologically relevant endogenous conditions, and validating the effects of different mutants, will be important to convince the experts and boost the value of the story.

Please be sure to address/respond to all concerns of the referees in full in a point-by-point response and highlight all changes in the revised manuscript text file. If you have comments that are intended for editors only, please include those in a separate cover letter.

We expect to see your revised manuscript within 3-6 months. If you cannot send it within this time, please contact us to discuss an extension; we would still consider your revision, provided that no similar work has been accepted for publication at NSMB or published elsewhere.

Reporting Summary:

When submitting the revised version of your manuscript, please pay close attention to our [href="https://www.nature.com/nature-portfolio/editorial-policies/image-integrity">Digital Image Integrity Guidelines.](https://www.nature.com/nature-portfolio/editorial-policies/image-integrity) and to the following points below:

Data availability: this journal strongly supports public availability of data. All data used in accepted papers should be available via a public data repository, or alternatively, as Supplementary Information. If data can only be shared on request, please explain why in your Data Availability Statement, and also in the correspondence with your editor. Please note that for some data types, deposition in a public repository is mandatory - more information on our data deposition policies and available repositories can be found below: <https://www.nature.com/nature-research/editorial-policies/reporting-standards#availability-of-data>

[redacted]

Note: This URL links to your confidential home page and associated information about manuscripts you may have submitted, or that you are reviewing for us.

If you wish to forward this email to co-authors, please delete the link to your homepage.

Sincerely,

Dimitris Typas
Associate Editor
Nature Structural & Molecular Biology
ORCID: 0000-0002-8737-1319

Referee expertise:

Referee #1: Ubiquitin structural biology, SAXS, HDX-MS

Referee #2: Ubiquitylation (structural and biochemistry), HECTs

Referee #3: Ubiquitylation (structural and biochemistry)

Reviewers' Comments:

Reviewer #1:

Remarks to the Author:

Düring et al describe a detailed investigation of the structural organization and activity regulation of the HECT type E3 ligase HACE1. They solve the structure of the protein using CryoEM and show that it adopts a dimeric state in which an N-terminal helix of one subunit makes crucial contacts with the N-lobe of the other subunit. Mutating crucial residues within this helix render the protein monomeric. The importance of this study lies in the fact that this dimeric state is an auto-inhibited one. The authors can also provide a structural explanation for this autoinhibition as binding of the E2 enzyme is inhibited. The monomeric version of the E3 ligase shows a higher activity, providing a link between activity and oligomerization. Furthermore, the authors solve the structure of a complex of the E3 ligase with a substrate covalently bound to the active site Cys residue and provide evidence that explains the selectivity of HACE1.

The data and the interpretation are very robust as the authors provide in addition to the direct structures further biochemical and functional evidence for all their claims. This includes enzymatic assays for investigating the activity of the different states as well as SAXS measurements and HDX MS measurements for probing the dynamic state of the dimer and the monomer. All claims are further supported by mutational analysis. Overall, this is a very detailed study of a very important and significant structural problem. The approaches chosen are valid, conclusions are robust and the text is clearly written.

I have only 2 minor suggestions.

1) In Supplementary figure S17 the ternary complex including the E3 charged with Ub and the substrate is shown. I would suggest to move this figure to the main text as it shows a model of the active complex or at least discuss this complex in more detail in the main text, in particular the conformational compatibility of the suggested Ub position with the

structure of the E3 with covalently bound substrate.

2) The authors mention that HACE1 also interacts with RAB GTPases. Are these also potential substrates like the GTPase Rac1? Or do the structures and mutational studies explain why RAB proteins are not targeted by HACE1?

Reviewer #2:

Remarks to the Author:

Ubiquitylation is an essential posttranslational modification that is carried out by specific E3 ubiquitin ligases that recruit substrates and catalyze ubiquitin transfer. The first class of E3 ligases were so-called HECT enzymes that possess a catalytic Cys residue that is charged with ubiquitin prior to transfer to the substrate. Only few HECT-family E3 ligases have been characterized in structural detail as full-length proteins, which leaves the mechanism of HECT-dependent ubiquitylation incompletely understood. Here, Düring, Wolter and colleagues provide cryo-EM structures of the full-length HECT E3 ligase HACE1, an enzyme that has roles in targeting the GTPase RAC1 for ubiquitylation but otherwise remains fairly poorly understood. They also show structures of HACE1 bound to RAC1. While on the lower end of high-resolution, their structures are informative and provide important starting points for analyzing the mechanism and regulation of HACE1. This work could therefore become an important steppingstone towards understanding important aspects of HECT-dependent ubiquitylation.

The authors focus on their observation that free HACE1 was found to be dimeric, which based on structural and their biochemical data inhibits activity. This could be interesting, but as described below, without careful studies of endogenous protein in cells it remains possible that dimerization is not physiologically relevant. In a similar manner, while they appear to identify the substrate binding interface in HACE1, they do not test whether introducing mutations at these sites confers phenotypes onto cells. Thus, while the structural work is very interesting, it remains to be seen whether it reflects important regulatory circuits in cells. More cell biology is clearly needed, and I suggest a few approaches below that could help the authors improve their study.

A similar cryo-EM structure of autoinhibited dimeric HACE1 was recently published (PMID 37537642). As this paper came out after the manuscript here had been submitted, I do not think it should be used as an argument against publication. However, this study showed a mixture of monomers and dimers, which again raises the relevance as to whether the dimeric, inhibited state of HACE1 is relevant without overexpression and in the presence of natural binding partners. This underscores that it is required to probe for the importance of the dimeric state using endogenous proteins (i.e. without overexpression).

Major points:

1. Because dimerization of HACE1 is a critical aspect of this manuscript, I am concerned that it was only documented in vitro with purified proteins or in cells without quantification and only upon overexpression. It would be important to show that endogenous HACE1 dimerizes in cells, and it would be critical to determine the extent of dimerization (i.e. is the proposed regulatory mechanism meaningful in cells?). This needs to be done without overexpression (the experiment in Fig. 2E is not sufficient in this case), as binding partners of HACE1 might very well determine how much of the protein is caught in an

inactive dimeric state. The authors could accomplish this by deleting the amino-terminal N-helix or introducing L11D or L15D mutations at the endogenous loci of HACE1 and compare size exclusion profiles in cell lysates to those of wildtype cells.

2. To really draw conclusions about substrate-modification through HACE1 and its regulation by dimerization, RAC1 should be directly visualized. The experiment in Fig. 2c, for example, looks at formation of ubiquitin chains, not RAC1 ubiquitylation, as it was ubiquitin that was fluorescently modified. Without directly visualizing RAC1, it is impossible to judge how much RAC1 was ubiquitylated and thus whether regulation by dimerization is meaningful (i.e. if only 1% of RAC1 is modified after the dimer interface has been disrupted, it is not meaningful). Along the same lines, it is essential to study the effect of such mutations in cells, where multiple substrates and/or regulators might compete for access to HACE1. The authors studied effects on dimer mutations on overexpressed HACE1 and overexpressed RAC1 in cells, but again, this is not sufficient, because dimerization plays a regulatory role and hence must be investigated at the endogenous level.

3. As with dimer interface mutants, the study of potential phosphorylation sites suffers from overexpression. Important information, such as extent of phosphorylation of the endogenous protein, is not provided. Whether and when such sites are phosphorylated is not discussed. Thus, it is difficult to assess the relevance of Figure 3.

4. To document the importance of HACE1 residues that are modelled to bind the E2, it would be better to introduce mutations (I694D, L704D etc) into the background of an activated delta-N HACE1 construct.

5. The structure of HACE1 bound to RAC1 is nice and at least initially characterized in vitro. However, it would very much improve the relevance of this study if consequences of mutations at the interface are shown in cells. This should be done at the endogenous level, not upon overexpression, by introducing HACE1 mutations identified in the structure into the endogenous HACE1 locus of cells. Potential phenotypes of these mutations should then be documented.

Minor points:

1. I would strongly suggest that the authors re-write the third paragraph of the introduction, which describes the biological functions of HACE1. First, they basically cite every manuscript that describes a potential biological function of this E3 ligase, without critically assessing the strength of the presented data. I believe it is fair to say that HACE1 ubiquitylates RAC1 and that its mutation or depletion leads to neurodevelopmental phenotypes and disease in frogs and humans. Even for RAC1 ubiquitylation, however, the authors need to distinguish between primary effects and potential secondary consequences that might be dependent on cell type, cell culture conditions, etc, i.e. conditions that are not really important in organisms. I would suggest that the authors suggest that the authors discuss HACE1 biology much more carefully. Second, they state that HACE1 can assemble multiple ubiquitin chain topologies, dependent on "context-dependent plasticity of its interactions with substrates and Ub". This likely reflects incomplete reconstitution of HACE1 activity in vitro, and substrate-binding should not have much of an effect on ubiquitin chain linkage. Again, it would have been helpful to evaluate existing papers a bit more carefully. It does not hurt the paper if it is built on trustworthy data.

2. While the authors focus on full-length HECT E3 ligases, HUWE1 and UBR5, that were recently characterized structurally, they should also discuss the groundbreaking work of Brenda Schulman in this field. Despite being done in yeast, it would be prudent to put more focus on this work in their discussion.

Reviewer #3:

Remarks to the Author:

In the manuscript "Structural mechanisms of autoinhibition and substrate recognition by the ubiquitin ligase HACE1", Düring et al. describe the cryo-EM structures of the full-length HECT E3 ubiquitin ligase HACE1 and of a complex of a truncated version of HACE1 (HACE1 Δ N) with its substrate RAC1. The authors support and validate their cryo-EM structures with complementary in solution techniques (SAXS, HDX-MS) and functional studies using purified recombinant proteins and overexpression in cell lines.

Examples of structures of full-length E3 ligases are sparse in the literature, as are structures of E3 ligase/substrate complexes. This work not only reports these structures for HACE1 (only the 3rd full-length HECT E3 ligase structure reported) but importantly, the structures, supported by biophysical and functional assays, reveal exciting mechanistic details of HACE1 autoinhibition and regulation. The authors find that HACE1 forms a homodimer where the N-terminal helix and neighbouring ankyrin repeats interact with a region in the C-terminal HECT domain, thereby preventing E2-Ub binding and Ub transfer from the E2 to HACE1. By truncating the N-terminal helix in HACE1 Δ N, the authors can disrupt the HACE1 dimer, leading to an active HACE1 monomer that is able to autoubiquitinate and ubiquitinate the substrate RAC1. This mechanism is further supported by functionally testing point mutations in the interaction sites. The authors suggest based on previously identified phosphorylation sites that phosphorylation in the N-terminal helix may regulate HACE1 dimerisation and activation. This notion is supported by introducing phosphomimetic mutations. This would be a very interesting regulator mechanism, which is somewhat underdeveloped in the manuscript.

The authors use a comprehensive integrated structural biology approach that will be of interest to many readers of the journal. Further, the mechanism-based chemical crosslinking method used to capture the E3/substrate complex is an innovative approach that will be appealing for other researchers in the field.

In summary, this is an excellent study that is well-written with clear and detailed figures. I only have few suggestions and requests for clarification before I can fully recommend publication.

Does the N-helix or phosphorylation of the N-helix have a function in recruiting or ubiquitinating RAC1? In Fig. 3C, it appears that the phosphomimetic mutants are more active in ubiquitinating RAC1 than the Δ N mutant (stronger fluorescent signal between 55 and 100 kDa). This seems to be specific for RAC1, as autoubiquitination and free chain formation are even faster in the Δ N mutant compared to the phosphomimetic mutants (Fig. 3D). The HACE1 Δ N-RAC1 structure supports this idea as RAC1 binds to the HACE1 ankyrin repeats which are close to the HACE1 N-terminus and therefore the N-helix in the full-length protein. Can the authors test this (e.g. using HDX-MS) using other mutants that disrupt the dimer and induce activity, e.g. one of the phosphomimetic mutations,

mutations in the small wing of the HECT N-lobe or in the N-terminal ankyrin repeats?

The authors identify the putative phosphorylation sites in the HACE1 N-helix from the PhosphoSitePlus database. Are these phosphorylation sites conserved in other organisms? This would further strengthen the notion that HACE1 activity is regulated by phosphorylation of these sites.

The authors identify three key interacting regions in the HACE1 dimer, the N-helix, the three N-terminal ankyrin repeats and the small wing of the HECT N-lobe, and they very much focus on the N-helix and partly the small wing of the HECT N-lobe. However, some of the strongest HDX differences between monomeric and dimeric HACE1 are seen in the ankyrin repeats (Fig 1G, S5). The authors should validate the importance of these interactions using mutations and functional assays as performed for the other interacting regions.

In the discussion (paragraph starting p. 13, l. 13), the authors outline that their crosslinking strategy may be suitable to crosslink other catalytic cysteine-dependent E3 ligases with a lysine residue in the E3s' substrate. I can clearly see the value of this strategy for other HECT and RBR E3 ligases, but the examples of MYCBP2 and RNF213 are not appropriate, as these ligases do not ubiquitinate lysine residues but hydroxyl groups on Ser/Thr residues and LPS, respectively. This should be clarified in the text.

Minor:

Fig 1G: I find some of the colours difficult to distinguish. The worst is the dark blue for < -20 % and the black for no peptides.

Figs. S4, S5, S6, S10: It would help to orient the reader in the HDX-MS difference plots of the different HACE1 domains were indicated, or at least the key regions with the major differences labelled.

Fig S19, legend: "HACE in grey" should be "HUWE1 in grey".

Author Rebuttal to Initial comments

Dear Reviewers,

we would like to thank you very much for carefully evaluating our manuscript, the insightful suggestions, and the unanimously positive feedback. We were delighted that all three Reviewers recognized the importance of our findings and provided such helpful input. Building on these inspiring suggestions, we feel that our manuscript has further improved.

In particular, the revised version of our manuscript includes:

- (i) An additional figure, showing **a donor Ub molecule modeled into our cryo-EM structure** of the HACE1 dimer (new Fig. 7b, requested by Reviewer 1).
- (ii) An additional discussion of the **interplay between HACE1 and RAB proteins** (page 15, requested by Reviewer 1).
- (iii) An additional figure, revealing that **endogenous HACE1 dimerizes in brain homogenates** from mice and rats, **where the dimer is the predominant form** (new Fig. 3a, requested by Reviewer 2).
- (iv) Additional co-IPs with **stably transfected cell lines expressing near-endogenous levels of HACE1 variants**, confirming that **dimerization depends on the N-helix** (new Fig. S8h, in response to Reviewer 2).
- (v) A clarification that **RAC1 is quantitatively ubiquitinated by active HACE1** in our *in-vitro* assays, as shown by existing Coomassie-stained images of all assays (requested by Reviewer 2).
- (vi) **Single-turnover E2 discharge and multi-turnover RAC1 ubiquitination assays with three additional HACE1 variants** (Δ N I694D, Δ N L704D, and Δ N L706D), showing that mutations in the conserved E2 binding site disrupt productive interactions of activated, monomeric HACE1 with the E2 (new Fig. 4e and new Fig. S11e,f, requested by Reviewer 2).
- (vii) **Cell-based assays with five HACE1 variants** to corroborate and expand our structure-guided functional validation of the HACE1-RAC1 interface (new Fig. 6i, in response to Reviewer 2).
- (viii) Adaptations to the introduction and discussion sections (page 4,14-16, requested by Reviewer 2).
- (ix) **Additional HDX-MS analyses of two dimerization-deficient HACE1 variants** (S14E and I694D) +/- RAC1, showing that the N-helix of HACE1 does not contribute to RAC1 binding (new Fig. S18 and S19, as requested by Reviewer 3).
- (x) **Sequence alignments of HACE1** and implications for regulation (new Fig. S9a; p.15) (requested by Reviewer 3).
- (xi) Mass photometry and enzymatic analyses of **three additional HACE1 variants** (Y32A, Q42A, and R44A), demonstrating that the ankyrin repeat 1 of HACE1 does not significantly contribute to dimerization (new Fig. S8d-g, as requested by Reviewer 3).
- (xii) Adaptions to the text and figures, addressing all minor requests by Reviewer 3.

Please find our detailed responses below.

Many thanks again for your efforts and best wishes,

-Sonja Lorenz

Reviewers' Comments:

Reviewer #1:

Remarks to the Author:

Düring et al describe a detailed investigation of the structural organization and activity regulation of the HECT type E3 ligase HACE1. They solve the structure of the protein using CryoEM and show that it adopts a dimeric state in which an N-terminal helix of one subunit makes crucial contacts with the N-lobe of the other subunit. Mutating crucial residues within this helix render the protein monomeric. The importance of this study lies in the fact that this dimeric state is an auto-inhibited one. The authors can also provide a structural explanation for this autoinhibition as binding of the E2 enzyme is inhibited. The monomeric version of the E3 ligase shows a higher activity, providing a link between activity and oligomerization. Furthermore, the authors solve the structure of a complex of the E3 ligase with a substrate covalently bound to the active site Cys residue and provide evidence that explains the selectivity of HACE1.

The data and the interpretation are very robust as the authors provide in addition to the direct structures further biochemical and functional evidence for all their claims. This includes enzymatic assays for investigating the activity of the different states as well as SAXS measurements and HDX MS measurements for probing the dynamic state of the dimer and the monomer. All claims are further supported by mutational analysis.

Overall, this is a very detailed study of a very important and significant structural problem. The approaches chosen are valid, conclusions are robust and the text is clearly written.

I have only 2 minor suggestions.

1) In Supplementary figure S17 the ternary complex including the E3 charged with Ub and the substrate is shown. I would suggest to move this figure to the main text as it shows a model of the active complex or at least discuss this complex in more detail in the main text, in particular the conformational compatibility of the suggested Ub position with the structure of the E3 with covalently bound substrate.

Thank you for highlighting the compatibility/relevance of donor Ub binding in the context of the RAC1-bound HACE1 active state that we identified. We added a main figure panel (new Fig. 7b), showing the donor Ub modeled into our cryo-EM structure of the RAC1-bound HACE1 monomer and comment on it:

“Notably, this complex is compatible with the conserved binding mode of the HACE1 C-lobe to the donor Ub (Fig. 7b), which further contributes to stabilizing the L-conformation^{47,48,63,64}.”
(page 15, line 37)

The Ub-containing AF2 model looks highly similar to this composite structural model and also remains part of the manuscript (now Fig. S15).

2) The authors mention that HACE1 also interacts with RAB GTPases. Are these also potential substrates like the GTPase Rac1? Or do the structures and mutational studies explain why RAB proteins are not targeted by HACE1?

This is a very interesting point that is still incompletely understood. We now comment on it in the discussion:

“As for RAC1, HACE1 interacts with RAB proteins in a GTP-dependent manner. Interestingly, RAB11 was reported to be ubiquitinated by HACE1 at a site that is homologous to Lys 147 of RAC1⁷². This suggests that HACE1 may allow for a degree of redundancy in the recognition of small GTPases as substrates – at least upon overexpression. Future studies are required to dissect the structural and physiological parameters underlying the selective recognition of RAB proteins by HACE1 and distinguish enzyme-substrate-type interactions from regulatory ones.” **(page 15, line 10)**

Reviewer #2:

Remarks to the Author:

Ubiquitylation is an essential posttranslational modification that is carried out by specific E3 ubiquitin ligases that recruit substrates and catalyze ubiquitin transfer. The first class of E3 ligases were so-called HECT enzymes that possess a catalytic Cys residue that is charged with ubiquitin prior to transfer to the substrate. Only few HECT-family E3 ligases have been characterized in structural detail as full-length proteins, which leaves the mechanism of HECT-dependent ubiquitylation incompletely understood. Here, Düring, Wolter and colleagues provide cryo-EM structures of the full-length HECT E3 ligase HACE1, an enzyme that has roles in targeting the GTPase RAC1 for ubiquitylation but otherwise remains fairly poorly understood. They also show structures of HACE1 bound to RAC1. **While on the lower end of high-resolution, their structures are informative and provide important starting points for analyzing the mechanism and regulation of HACE1. This work could therefore become an important steppingstone towards understanding important aspects of HECT-dependent ubiquitylation.**

The authors focus on their observation that free HACE1 was found to be dimeric, which based on structural and their biochemical data inhibits activity. This could be interesting, but as described below, without careful studies of endogenous protein in cells it remains possible that dimerization is not physiologically relevant. In a similar manner, while they appear to identify the substrate binding interface in HACE1, they do not test whether introducing mutations at these sites confers phenotypes onto cells. Thus, while the structural work is very interesting, it remains to be seen whether it reflects important regulatory circuits in cells. More cell biology is clearly needed, and I suggest a few approaches below that could help the authors improve their study.

A similar cryo-EM structure of autoinhibited dimeric HACE1 was recently published (PMID 37537642). As this paper came out after the manuscript here had been submitted, I do not think it should be used as an argument against publication. However, this study showed a mixture of monomers and dimers, which again raises the relevance as to whether the dimeric, inhibited state of HACE1 is relevant without overexpression and in the presence of natural binding partners. This underscores that it is required to probe for the importance of the dimeric state using endogenous proteins (i.e. without overexpression).

We thank the Reviewer for bringing up this interesting point. Like Singh et al. (Adv. Sci., 2023), we noticed monomer-like particles on cryo-EM grids and attribute them to the dimer suffering damage at the air-water interface during freezing. Additional density is observed on these apparent monomers, which, in our hands, likely represents parts of a flanking HECT domain of a broken dimeric particle. Singh et al. assign the extra density to a putative uncleaved GST-tag present in their system. Whichever assignment applies, it is clear from our analyses by SEC, mass photometry, and SAXS that purified, full-length HACE1 WT is predominantly dimeric (Fig. 1a,b,e,f; Fig. S3); monomers are not detected in solution. We included this point into the discussion:

“Our in-vitro analyses reveal that purified HACE1 FL is predominantly dimeric in solution, with no monomers being detected by various biophysical techniques. Similar to Singh et al.⁶⁶, however, we observe monomer-like particles on the cryo-EM grids, which we attribute to dimers likely being damaged at the air-water interface during freezing.” (page 14, line 27)

Importantly, in the revised version our manuscript, we now provide evidence that the dimeric state of HACE1 occurs at endogenous levels in brain homogenates, as discussed below.

Major points:

1. Because dimerization of HACE1 is a critical aspect of this manuscript, I am concerned that it was only documented in vitro with purified proteins or in cells without quantification and only upon overexpression. It would be important to show that endogenous HACE1 dimerizes in cells, and it would be critical to determine the extent of dimerization (i.e. is the proposed regulatory mechanism meaningful in cells?). This needs to be done without overexpression (the experiment in Fig. 2E is not sufficient in this case), as binding partners of HACE1 might very well determine how much of the protein is caught in an inactive dimeric state. The authors could accomplish this by deleting the amino-terminal N-helix or introducing L11D or L15D mutations at the endogenous loci of HACE1 and compare size exclusion profiles in cell lysates to those of wildtype cells.

Thank you for raising this important aspect, which we successfully addressed in an endogenous context, without overexpression: we analyzed brain homogenates from mice and rats, in which the amino acid sequence of HACE1 is ~ 97% and the dimerization region 100% conserved compared to human. Native PAGE analyses reveal that endogenous HACE1 migrates similarly to recombinant dimeric HACE1 at a molecular weight of ~ 200 kDa and distinct from the recombinant monomeric form (new Fig. 3a). No monomer is observed for endogenous HACE1 under these conditions. The added results paragraph reads:

“We next sought to test whether the HACE1 dimer is detectable in a physiological setting without any overexpression. Due to the significance of HACE1 in neuronal homeostasis^{23,34–37}, we focused on mouse and rat-derived brain tissues. Notably, HACE1 FL is highly conserved between human, mouse and rat, as reflected by ~ 97% overall amino acid sequence identity and 100% identity within the dimerization region (Fig. S9a). The brain homogenates were centrifuged at low speed, the resulting supernatant was separated from the pellet and analyzed by native PAGE, followed by immunoblotting (IB) against HACE1. Intriguingly, endogenous HACE1 migrates at ~ 200 kDa, similar to the recombinant HACE1 FL dimer and distinct from the recombinant Δ N monomer (Fig. 3a). These experiments suggest that HACE1 may dimerize at endogenous levels and that the dimer represents the predominant form in the brain homogenate fractions analyzed here. This raises the question of which mechanisms regulate the oligomerization and activity of HACE1 in different biological contexts.” (page 8, line 33)

That we observe the HACE1 dimer both at endogenous levels and upon overexpression in cells is consistent with the fact that it is formed across a wide concentration range from low-nanomolar to micromolar in vitro (as monitored by analytical SEC, mass photometry, SAXS, cryo-EM, HDX-MS, and enzyme assays). Likewise, we show that structure-based mutations efficiently disrupt dimerization across the same wide concentration range. Together, this makes it very likely that the HACE1 dimer that we characterized corresponds to the dimeric form detected for endogenous HACE1. Nevertheless, we further addressed this point by establishing a system allowing us to probe the effect of dimer-disruptive mutations without strong overexpression of HACE1. We generated stably transfected HEK293 cell lines for the inducible expression of the relevant HACE1 variants (FL, Δ N, and L15D). In the absence of the inducer, these cell lines have a low level of leaky expression that is rather close to that of endogenous HACE1 (< 2-fold excess). Indeed, co-IP experiments performed using this system recapitulate our finding that the self-association of HACE1 depends on its N-terminal helix. The new data were summarized as follows:

“To assess whether HACE1 also self-associates in an N-helix-dependent manner at concentrations that are closer to the endogenous levels of this ligase, we employed stably transfected HEK293 Flp-In T-REx (Thermo Fisher Scientific) cell lines, expressing GFP-tagged HACE1 FL, L15D, and Δ N, respectively, from a tetracycline-inducible promoter. In these cells, in the absence of the inducer, a small amount of ‘leaky’ expression occurs that is only slightly higher than the level of endogenous HACE1 (Fig. S8h). Capitalizing on this system, we performed co-IP experiments, confirming that HACE1 FL self-associates and that this interaction is sensitive to the L15D substitution or truncation of the N-helix (Fig. S8h). In

line with our in-vitro results, the N-helix-dependent self-association of HACE1 FL can thus be observed across a wide range of concentrations in cells.” (page 8, line 21)

2. To really draw conclusions about substrate-modification through HACE1 and its regulation by dimerization, RAC1 should be directly visualized. The experiment in Fig. 2c, for example, looks at formation of ubiquitin chains, not RAC1 ubiquitylation, as it was ubiquitin that was fluorescently modified. Without directly visualizing RAC1, it is impossible to judge how much RAC1 was ubiquitylated and thus whether regulation by dimerization is meaningful (i.e. if only 1% of RAC1 is modified after the dimer interface has been disrupted, it is not meaningful). Along the same lines, it is essential to study the effect of such mutations in cells, where multiple substrates and/or regulators might compete for access to HACE1. The authors studied effects on dimer mutations on overexpressed HACE1 and overexpressed RAC1 in cells, but again, this is not sufficient, because dimerization plays a regulatory role and hence must be investigated at the endogenous level.

We appreciate this comment, but would like to point out that we show Coomassie stains for all gel-based activity assays, in addition to the fluorescence images (Fig. S7, S8, S9, S11, and S20). The stained gels provide a direct visualization of unmodified and modified RAC1. They demonstrate that RAC1 is quantitatively ubiquitinated by activated HACE1, as seen from the disappearance of the unmodified RAC1 band. The structure-based mutations we introduced are thus effective. We added a comment at the point where the first such assay is introduced: *“Note that the extent of HACE1-driven RAC1 modification can be evaluated from Fig. S7b.” (page 7, line 24).*

As described above, we now provide evidence for dimerization of endogenous HACE1 in brain homogenates (new Fig. 3a) and validated the dimerization interface by mutational analyses in a system with very limited overexpression (new Fig. S8h). To additionally validate our structural model by endogenous mutations is an interesting suggestion, but beyond the scope of this study.

3. As with dimer interface mutants, the study of potential phosphorylation sites suffers from overexpression. Important information, such as extent of phosphorylation of the endogenous protein, is not provided. Whether and when such sites are phosphorylated is not discussed. Thus, it is difficult to assess the relevance of Figure 3.

This section (now entitled “Phosphorylation provides a possible mechanism to regulate HACE1 dimerization”) provides a structure-inspired suggestion of how the dimer interface may be regulated. Based on the fact that a cluster of physiological phosphorylation sites is seen at the dimer interface in our cryo-EM structure, we propose that phosphorylation may be a regulatory mechanism and corroborate this possibility with phospho-mimetic mutations. We took care not to make any unsupported claims and state the interesting open questions in the discussion section. We have included the Reviewer’s points explicitly in this discussion:

“We further provide evidence that HACE1 self-associates via the N-helix upon limited overexpression in cells and that endogenous HACE1 can form a dimer in animal-derived brain homogenate fractions. This raises the important question of which physiological factors modulate the oligomerization state of HACE1 according to cellular demand. While answering this question requires future, cell biological studies of endogenous HACE1, we propose that the phosphorylation of linchpin sites in the N-terminal region of the ligase may stabilize its monomeric, active state. These sites can be phosphorylated in the cell⁵⁶, but the context and extent of phosphorylation, and the kinase(s) responsible remain to be identified. Possible entry points may be provided by recent interactomic analyses that detected a number of candidate HACE1-binding kinases⁷¹. It is also conceivable that the catalytic activity of HACE1 is regulated by yet unknown interactors of the amphipathic N-helix and/or the recruitment of the ligase to cellular membranes...” (page 14, line 31)

4. To document the importance of HACE1 residues that are modelled to bind the E2, it would be better to introduce mutations (I694D, L704D etc) into the background of an activated delta-N HACE1 construct.

Thank you for this insightful point. As requested, we generated and analyzed the suggested three mutations (I694D, L704D, and L806D) in the context of HACE1 Δ N (new Fig. 4e; Fig. S11e,f). The results confirm our previous conclusion: the mutations I694E, L704D and L706D disrupt E2-mediated Ub transfer to HACE1, also in the activated, monomeric state. The adapted text reads as follows:

“As expected, neither of the variants was able to promote Ub discharge from the E2 (Fig. 4d; Fig. S11b), HACE1 autoubiquitination, or RAC1 ubiquitination in the context of HACE1 FL (Fig. S11c,d). Importantly, the same mutations also disrupted activity when introduced into the Δ N context (Fig. 4e; Fig. S11e,f)...” (**page 10, line 14**)

5. The structure of HACE1 bound to RAC1 is nice and at least initially characterized in vitro. However, it would very much improve the relevance of this study if consequences of mutations at the interface are shown in cells. This should be done at the endogenous level, not upon overexpression, by introducing HACE1 mutations identified in the structure into the endogenous HACE1 locus of cells. Potential phenotypes of these mutations should then be documented.

We thank the Reviewer for recognizing the importance of the HACE1-RAC1 structure. In response to this comment, we have expanded our structure-guided validation of the RAC1 binding site in HACE1 to a cellular context. We analyzed the panel of interface mutations (R107A, N108A, V140L, Q173A and N174A) in cell-based RAC1-ubiquitination assays using transient transfections. The results recapitulate the defects in RAC1 ubiquitination that we had previously validated *in vitro*. The new data are included as Fig. 6i and accompanied by the following text:

“Encouragingly, the panel of structure-based mutations in zones I to III also perturb the activity of HACE1 towards RAC1 Q61L upon overexpression in HeLa cells (Fig. 6i). R107A, Q173A and N174A appear more disruptive than N108A and V140L, recapitulating the relative effects observed *in vitro*.” (**page 13, line 30**)

We agree that follow-up analyses of HACE1-RAC1-related phenotypes at endogenous levels would be interesting and will include them into our future work. However, they are beyond the scope of the structural mechanisms this study focuses on.

Minor points:

1. I would strongly suggest that the authors re-write the third paragraph of the introduction, which describes the biological functions of HACE1. First, they basically cite every manuscript that describes a potential biological function of this E3 ligase, without critically assessing the strength of the presented data. I believe it is fair to say that HACE1 ubiquitylates RAC1 and that its mutation or depletion leads to neurodevelopmental phenotypes and disease in frogs and humans. Even for RAC1 ubiquitylation, however, the authors need to distinguish between primary effects and potential secondary consequences that might be dependent on cell type, cell culture conditions, etc, i.e. conditions that are not really important in organisms. I would suggest that the authors suggest that the authors discuss HACE1 biology much more carefully.

We thank the Reviewer for this input and adapted the respective paragraph. We now provide a compact overview of the functions and substrates of HACE1, as reported in the literature, while explicitly highlighting system-dependencies, both in the general section and below, when introducing RAC1 as a HACE1 substrate. The rephrased text reads as follows:

” HACE1 has been implicated in redox homeostasis^{22–24} and membrane dynamics, including cell migration and adhesion^{25–27}, autophagy^{28,29}, and Golgi turnover³⁰. Expression of HACE1 was reported to confer protection against haemodynamic stress in the heart²⁹ and tumorigenesis^{31–33}, while loss-of-function mutations or depletion of HACE1 lead to

neurodevelopmental deficiencies and disease in frogs and humans^{23,34–37}. In several of these settings, phenotypes were linked with loss of the catalytic activity of the ligase^{22,24,25,28,31,38}. It is important to note, however, that the precise pathways underlying the reported, diverse roles of HACE1 are likely system-dependent and incompletely understood. Only few substrates of HACE1 have been identified, including the selective autophagy receptor optineurin (OPTN)²⁸, the Golgi t-SNARE syntaxin 5³⁹, and the TNFR1-associated adaptor TRAF2³⁸. The most established substrate is the multifunctional small GTPase RAC1^{22,25–27,40}.“ (page 3, line 36)

...

“The ensuing selective downregulation of active RAC1 was reported to restrict a range of RAC1-related cellular functions in a context-dependent manner. Interestingly, neuropathologic features upon HACE1 deficiency, as seen in SPPRS (spastic paraplegia and psychomotor retardation with or without seizures) patients, are also accompanied by elevated levels of active RAC1 and induction of RAC1-dependent signaling pathways³⁵. Moreover, the tumor suppressive functions of HACE1 in lung cancer were associated with the suppression of oncogenic RAC-family GTPase activities³³.“ (page 4, line 19)

Second, they state that HACE1 can assemble multiple ubiquitin chain topologies, dependent on “context-dependent plasticity of its interactions with substrates and Ub”. This likely reflects incomplete reconstitution of HACE1 activity in vitro, and substrate-binding should not have much of an effect on ubiquitin chain linkage. Again, it would have been helpful to evaluate existing papers a bit more carefully. It does not hurt the paper if it is built on trustworthy data.

We have re-phrased this sentence to avoid ambiguities:

“Different Ub modifications types have been detected for HACE1 substrates^{25,28,38,39,41}, yet how the ligase assembles these modifications and which determinants confer specificity has not been studied at a structural level. To gain insight into the mechanisms of substrate recognition by HACE1, we focused on the interaction with RAC1 ...” (page 4, line 9).

Please note that we included an additional citation here (Tortola et al, Cell Reports, 2018) that we had unfortunately forgotten to list at this position. As we consider it possible that HACE1 forms or impacts the formation of different chain topologies in a context- and/or study system-dependent manner, we feel that it is important to offer readers a comprehensive list of references on this interesting topic.

2. While the authors focus on full-length HECT E3 ligases, HUWE1 and UBR5, that were recently characterized structurally, they should also discuss the groundbreaking work of Brenda Schulman in this field. Despite being done in yeast, it would be prudent to put more focus on this work in their discussion.

We fully recognize the landmark work of Brenda Schulman and cited seven of her primary research papers and two reviews, as applicable to our study, in the original version of the manuscript (including her work on yeast Rsp5 (Kamadurai et al., 2013)). We understand that the Reviewer’s comment refers to the discussion section in particular. Here, we focus on comparing our findings to other structurally characterized full-length HECT-type ligases and thus specifically cited the recent work of the Schulman lab on UBR5 (previously cited as a preprint, now as the published paper (Hehl et al.; citation #48)). However, it is important to us that no misconceptions arise, so we followed the Reviewer’s advice and added the following sentence to the discussion:

“In this context, it is important to note that the conformational cycle of the HECT domain and its interactions with the E2 and donor Ub, as established by pioneering work on HECT ligase fragments^{58,65,75,76}, are conserved within the recently characterized Ub transfer complexes of the full-length enzymes^{48,49}.“ (page 16, line 28)

Reviewer #3:

Remarks to the Author:

In the manuscript “Structural mechanisms of autoinhibition and substrate recognition by the ubiquitin ligase HACE1”, Düring et al. describe the cryo-EM structures of the full-length HECT E3 ubiquitin ligase HACE1 and of a complex of a truncated version of HACE1 (HACE1 Δ N) with its substrate RAC1. The authors support and validate their cryo-EM structures with complementary in solution techniques (SAXS, HDX-MS) and functional studies using purified recombinant proteins and overexpression in cell lines.

Examples of structures of full-length E3 ligases are sparse in the literature, as are structures of E3 ligase/substrate complexes.

This work not only reports these structures for HACE1 (only the 3rd full-length HECT E3 ligase structure reported) but importantly, the structures, supported by biophysical and functional assays, reveal exciting mechanistic details of HACE1 autoinhibition and regulation. The authors find that HACE1 forms a homodimer where the N-terminal helix and neighbouring ankyrin repeats interact with a region in the C-terminal HECT domain, thereby preventing E2-Ub binding and Ub transfer from the E2 to HACE1. By truncating the N-terminal helix in HACE1 Δ N, the authors can disrupt the HACE1 dimer, leading to an active HACE1 monomer that is able to autoubiquitinate and ubiquitinate the substrate RAC1. **This mechanism is further supported by functionally testing point mutations in the interaction sites. The authors suggest based on previously identified phosphorylation sites that phosphorylation in the N-terminal helix may regulate HACE1 dimerisation and activation. This notion is supported by introducing phosphomimetic mutations.** This would be a very interesting regulator mechanism, which is somewhat underdeveloped in the manuscript.

The authors use a comprehensive integrated structural biology approach that will be of interest to many readers of the journal. Further, the mechanism-based chemical crosslinking method used to capture the E3/substrate complex is an innovative approach that will be appealing for other researchers in the field.

In summary, this is an excellent study that is well-written with clear and detailed figures. I only have few suggestions and requests for clarification before I can fully recommend publication.

- Does the N-helix or phosphorylation of the N-helix have a function in recruiting or ubiquitinating RAC1? In Fig. 3C, it appears that the phosphomimetic mutants are more active in ubiquitinating RAC1 than the Δ N mutant (stronger fluorescent signal between 55 and 100 kDa). This seems to be specific for RAC1, as autoubiquitination and free chain formation are even faster in the Δ N mutant compared to the phosphomimetic mutants (Fig. 3D). The HACE1 Δ N-RAC1 structure supports this idea as RAC1 binds to the HACE1 ankyrin repeats which are close to the HACE1 N-terminus and therefore the N-helix in the full-length protein. Can the authors test this (e.g. using HDX-MS) using other mutants that disrupt the dimer and induce activity, e.g. one of the phosphomimetic mutations, mutations in the small wing of the HECT N-lobe or in the N-terminal ankyrin repeats?

To address this insightful comment, we performed additional HDX-MS analyses on the dimerization-deficient variants S14E (phospho-mimetic mutation in the N-helix) and I694D (disruption of E2 binding) and compared their HDX profiles in the absence and presence of RAC1 Q61L. These experiments do not provide any evidence of a participation of the N-helix in RAC1 recruitment (new Fig. S18, S19). However, they yield additional interesting insight: (i) they support the RAC1 binding mode we had previously characterized for the Δ N variant (Fig. S17), (ii) they indicate that the presence of RAC1 may promote the monomeric state of the S14E variant, and (iii) they reveal that the I694D mutation increases the conformational flexibility of the HECT N-lobe. We have summarized these new findings as follows:

“As our cryo-EM structure of the HACE1-RAC1 complex does not contain the N-helix, we used additional HDX experiments to interrogate the possibility that the N-helix may contribute to

RAC1 binding by full-length, monomerized HACE1. We compared the HDX profiles of the dimerization-deficient, full-length S14E and I694D variants, respectively, in the presence and absence of RAC1 Q61L (Fig. S18, S19). As expected, both variants consistently show RAC1-induced HDX reductions in the ankyrin repeats that resemble those seen upon RAC1 binding to the DN variant, indicative of a common binding mode. In addition, RAC1 enhances HDX near the N-terminus and in the N-lobe of the S14E variant, recapitulating the changes observed upon RAC1 binding to the HACE1 dimer (Fig. S5, S6, S10). This suggests that RAC1 binding may stabilize the monomeric state of this variant. In contrast, the I694D variant does not experience RAC1-induced HDX enhancements in these regions, which may indicate that it is fully monomeric. Instead, elevated HDX is observed in other areas of the HECT N-lobe, which we interpret as mutation-induced changes in the conformational dynamics of the inherently flexible HECT domain. Importantly, however, neither of the two monomerized, full-length HACE1 variants displays HDX changes indicative of an interaction between the N-helix and RAC1. This suggests that the binding mode seen in our cryo-EM structure of the HACE1-RAC1 complex and supported by common RAC1-induced HDX decreases in all tested dimerization-deficient variants (DN, S14E, and I694D), is independent of the N-helix.” (page 12, line 30)

- The authors identify the putative phosphorylation sites in the HACE1 N-helix from the PhosphoSitePlus database. Are these phosphorylation sites conserved in other organisms? This would further strengthen the notion that HACE1 activity is regulated by phosphorylation of these sites.

This is an interesting point. As shown below, full-length HACE1 is overall highly conserved among many vertebrates. It is thus not possible to interpret the conservation of the phosphorylation sites specifically. Note that the least conserved region (residues 383 to 444) corresponds to the loop 2 that is not resolved in our cryo-EM structure.

We included a sequence alignment of HACE1 from human, mouse, and rat into the revised manuscript (new Fig. S9a; comment on page 8, line 35), along with the analyses on mice and rat brain homogenates (new Fig. 3a).

Amino acid sequence alignment of full-length HACE1 from the indicated species, generated with PSI-BLAST and illustrated with Jalview, using Blossum 62 coloring.

We noticed, however, that public databases report two HACE1 isoforms that are N-terminally truncated by 34 residues and thus devoid of the N-helix. Based on our model, we expect these isoforms would be unable to adopt the dimeric autoinhibited state as characterized by us. However, any possible functions of the N-helix which may regulate the localization, activity or specificity of HACE1 would also be altered. In future studies, it will be interesting to explore whether such isoforms are expressed at physiologically relevant levels and to study the associated biological context and consequences. We included this point into the discussion: *“Finally, it is possible that HACE1 is regulated at the transcriptional level. Public databases⁷³ report several human HACE1 isoforms of which two lack the N-terminal 34 residues (NCBI reference sequence ID: NP_001308012.1 and NP_001337483.1), including the critical N-helix. It will be interesting to explore whether in certain contexts these isoforms are expressed at physiologically relevant levels and, if so, which distinct activities and specificities they may confer.”* (page 15, line 16).

- The authors identify three key interacting regions in the HACE1 dimer, the N-helix, the three N-terminal ankyrin repeats and the small wing of the HECT N-lobe, and they very much focus on the N-helix and partly the small wing of the HECT N-lobe. However, some of the strongest HDX differences between monomeric and dimeric HACE1 are seen in the ankyrin repeats (Fig 1G, S5). The authors should validate the importance of these interactions using mutations and functional assays as performed for the other interacting regions.

To address this point, we performed additional mass photometry analyses as well as RAC1 ubiquitination and E3 autoubiquitination assays to interrogate the effect of three structure-guided mutations within the first ankyrin repeat that makes peripheral, polar contacts at the dimer interface in our structure. Note that no relevant intermolecular contacts are contributed by the adjacent ankyrin repeats. The additional mutational analyses (new Fig. S8d-g) do not provide evidence of a significant contribution of ANK1 to dimerization. Hence, the monomerization-induced HDX enhancement that we observed in a portion of ANK1 and the adjacent ankyrin repeats is likely due to propagated changes in the conformational dynamics that originate from the N-helix. Similar conformational ‘sensitivity’/ fluctuations across ankyrin repeats have also been observed in other HDX-MS analyses (e.g., Goretzki et al., *Nature Commun.*, 2023; PMID: 37443299). The new text reads:

“Aside from the N-helix, the first ankyrin repeat (ANK1) contributes some intermolecular, polar contacts in our cryo-EM structure of the HACE1 dimer. To interrogate the significance of these contacts, we introduced alanine substitutions of Tyr 32, Gln 42, and Arg 44 (Fig. S8d). However, those did not markedly affect the dimerization propensity and activity of HACE1 (Fig. S8e-g). This suggests that ANK1 does not have a major role in HACE1 dimerization. Nevertheless, the N-terminal portion of ANK1 and adjacent ankyrin repeats experience enhanced HDX upon the disruption of the dimer (Fig. 1g; Fig. S5, S6). This likely reflects monomerization-induced changes in the conformational dynamics of HACE1 that originate from the N-helix and are propagated throughout the ankyrin repeats.” (page 7, line 35)

- In the discussion (paragraph starting p. 13, l. 13), the authors outline that their crosslinking strategy may be suitable to crosslink other catalytic cysteine-dependent E3 ligases with a lysine residue in the E3s’ substrate. I can clearly see the value of this strategy for other HECT and RBR E3 ligases, but the examples of MYCBP2 and RNF213 are not appropriate, as these ligases do not ubiquitinate lysine residues but hydroxyl groups on Ser/Thr residues and LPS, respectively. This should be clarified in the text.

Thank you very much for pointing this out. We have corrected the text accordingly:

“While the selectivity of this strategy is system-dependent, we envision it to be applicable to structural analyses of additional complexes of HECT-type ligases with substrates and other catalytic cysteine-dependent E3s that modify primary amino groups.” (page 15, line 31).

Minor:

- Fig 1G: I find some of the colours difficult to distinguish. The worst is the dark blue for < -20 % and the black for no peptides.

We thank the Reviewer for highlighting this aspect. We would like to clarify that Fig. 1g does not contain regions with “less HDX in ΔN ” and thus no shades of blue (see figure legend 1g: “no rigidification (shades of blue) occurs”). We had included the blue colors in the legend only for consistency with Fig. 5h. To avoid misunderstandings, we have now chosen a distinct color for regions without peptide coverage (beige), such that the blue colors, if present, are easier to spot.

- Figs. S4, S5, S6, S10: It would help to orient the reader in the HDX-MS difference plots of the different HACE1 domains were indicated, or at least the key regions with the major differences labelled.

Thank you for this suggestion. We have added full domain labels in all HDX-MS plots (Fig. S4, S5, S6, S10, S16, S17, S18, and S19).

- Fig S19, legend: “HACE in grey” should be “HUWE1 in grey”.

Thanks! Done.

Decision Letter, first revision:

Message: Our ref: NSMB-A48027A

18th Oct 2023

Dear Dr. Lorenz,

Thank you for submitting your revised manuscript "Structural mechanisms of autoinhibition and substrate recognition by the ubiquitin ligase HACE1" (NSMB-A48027A). It has now been seen by the original referees and their comments are below. The reviewers find that the paper has improved in revision, and therefore we'll be happy to accept it in principle in Nature Structural & Molecular Biology, pending minor revisions to satisfy the referees' final requests and to comply with our editorial and formatting guidelines.

We are now performing detailed checks on your paper and will send you a checklist detailing our editorial and formatting requirements in about two weeks. Please do not upload the final materials and make any revisions until you receive this additional information from us.

To facilitate our work at this stage, it is important that we have a copy of the main text as a word file. If you could please send along a word version of this file as soon as possible, we would greatly appreciate it; please make sure to copy the NSMB account (cc'ed above).

Sincerely,

Dimitris Typas
Associate Editor
Nature Structural & Molecular Biology
ORCID: 0000-0002-8737-1319

Reviewer #1 (Remarks to the Author):

The authors have addressed all my concerns.

Reviewer #2 (Remarks to the Author):

While I would have welcomed a more careful analysis of endogenous dimerization mutants using gene editing approaches, I believe that the additional experiments performed by the authors support their notion that (a) HACE1 dimerizes to a certain extent; (b) that dimerization causes autoinhibition; and (c) that substrates, such as RAC1, engage a monomeric E3 ligase for ubiquitylation. I am still not excited about the potential regulation

by phosphorylation, as the extent of such modification is not known even at a basic level. However, given the additional data, this manuscript represents significant progress in the structural understanding of HECT-family E3 ligases and I support publication.

Reviewer #3 (Remarks to the Author):

The authors have thoroughly addressed all my suggestions and requests with additional experiments and clarifications in the text. I now support publication of the revised manuscript.

Final Decision Letter:

Message 7th Dec 2023

:

Dear Dr. Lorenz,

We are now happy to accept your revised paper "Structural mechanisms of autoinhibition and substrate recognition by the ubiquitin ligase HACE1" for publication as an Article in Nature Structural & Molecular Biology.

Your paper will be published online soon after we receive proof corrections and will appear in print in the next available issue. You can find out your date of online publication by contacting the production team shortly after sending your proof corrections. Content is published online weekly on Mondays and Thursdays, and the embargo is set at 16:00 London time (GMT)/11:00 am US Eastern time (EST) on the day of publication. Now is the time to inform your Public Relations or Press Office about your paper, as they might be interested in promoting its publication. This will allow them time to prepare an accurate and satisfactory press release. Include your manuscript tracking number (NSMB-A48027B) and our journal name, which they will need when they contact our press office.

About one week before your paper is published online, we shall be distributing a press release to news organizations worldwide, which may very well include details of your work. We are happy for your institution or funding agency to prepare its own press release, but it must mention the embargo date and Nature Structural & Molecular Biology. If you or your Press Office have any enquiries in the meantime, please contact press@nature.com.

Please note that *Nature Structural & Molecular Biology* is a Transformative Journal (TJ). Authors may publish their research with us through the traditional subscription access route or make their paper immediately open access through payment of an article-processing charge (APC). Authors will not be required to make a final decision about access to their article until it has been accepted. Access to their article until it has been accepted.

<https://www.springernature.com/gp/open-research/transformative-journals>> Find out more about Transformative Journals

Authors may need to take specific actions to achieve compliance with funder and institutional open access mandates. If your research is supported by a funder that requires immediate open access (e.g. according to Plan S principles) then you should select the gold OA route, and we will direct you to the compliant route where possible. For authors selecting the subscription publication route, the journal's standard licensing terms will need to be accepted, including self-archiving policies. Those licensing terms will supersede any other terms that the author or any third party may assert apply to any version of the manuscript.

Sincerely,

Dimitris Typas
Associate Editor
Nature Structural & Molecular Biology
ORCID: 0000-0002-8737-1319